# Contraceptive use and needs among adolescent women aged 15–19: Regional and global estimates and projections from 1990 to 2030 from a Bayesian hierarchical modelling study

**Vladimíra Kantorová[1]\*, Mark C. Wheldon[1], Aisha N. Z. Dasgupta[1], Philipp Ueffing[1], Helena Cruz Castanheira[2]**

**1** Population Division, United Nations, Department of Economic and Social Affairs, New York, NY, United States of America, **2** United Nations, Economic Commission for Latin America and the Caribbean, Santiago de Chile, Chile

\* kantorova@un.org

## Abstract

Expanding access to contraception and ensuring that need for family planning is satisfied are essential for achieving universal access to reproductive healthcare services, as called for in the 2030 Agenda for Sustainable Development. To quantify the gaps that remain in meeting needs among adolescents, this study provides a harmonised data set and global estimates and projections of family planning indicators for adolescents aged 15–19 years. We compiled a comprehensive dataset of family-planning indicators among women aged 15–19 from 754 nationally representative surveys. We used a Bayesian hierarchical model with country-specific annual trends to estimate contraceptive prevalence and unmet need for family planning, with 95% uncertainty intervals (UIs), for 185 countries, taking into account changes in proportions married or in a union and differences in sexual activity among unmarried women across countries. Among 300 million women aged 15–19 years in 2019, 29.8 million (95% UI 24.6–41.7) use any contraception, and 15.0 million (95% UI 12.1–29.2) have unmet need for family planning. Population growth and the postponement of marriage influence trends in the absolute number of adolescents using contraception or experiencing unmet need. Large gaps remain in meeting family-planning needs among adolescents. The proportion of the need satisfied by modern methods, Sustainable Development Goals (SDG) indicator 3.7.1, was 59.2% (95% UI 44.8–67.2) globally among adolescents, lower compared to 75.7% (95% UI 73.2%–78.0%) among all women age 15–49 years. It was less than one half of adolescents in need in Western Asia and Northern Africa (38.7%, 95%UI = 20.9–56.5), Central and Southern Asia (43.5%, 95%UI = 36.6–52.3), and sub-Saharan Africa (45.6%, 95%UI = 42.2–49.0). The main limitations of the study are: (i) the uncertainty surrounding estimates for countries with limited or biased data is large; and (ii) underreporting of contraceptive use and needs is likely, especially among unmarried adolescents.

**Data Availability Statement:** All relevant data are within the manuscript and its Supporting Information files.

**Funding:** VK, MCW, PU, and ANZD received funding from the Bill and Melinda Gates Foundation at https://www.gatesfoundation.org/ (Grants Nos. OPP1183453 and OPP1110679). The funders had no role in study design, data collection and analysis, decision to publish, or preparation of the manuscript.

**Competing interests:** The authors have declared that no competing interests exist.

# Introduction

Many girls and young women do not have access to the information and services they need to promote their sexual and reproductive health and to fulfil their reproductive rights [1]. They also face challenges in accessing services due to financial barriers, provider bias, or concerns around confidentiality; once married, they may feel social pressure to have a baby soon after marriage [2,3]. It has been estimated that around half of all pregnancies among women aged 15–19 in developing regions in 2016 were unintended and that more than half of these ended in abortion, often under unsafe conditions [4]. In some countries, an adolescent pregnancy often precipitates marriage, whether or not the pregnancy was intended, forever changing the life course of the pregnant girl or young woman. Early childbearing is known to have long-term consequences for the mother's educational attainment, employment prospects and future health status, while also increasing her risk of poverty. For society, a widespread practice of early childbearing diminishes the chances of achieving gender equality. Globally in 2019, there were around 300 million girls and young women between the ages of 15 and 19 years. Many of these adolescents are now, or soon will be, sexually active. Almost 90% of them live in low- or middle-income countries [5].

Recently there have been calls to improve the evidence base on adolescent and young women's sexual and reproductive health status to inform family planning policies and programming, satisfy the monitoring requirements of the 2030 Agenda for Sustainable Development and ensure accountability at the local, national and global levels [6–9]. Within the monitoring framework adopted by Governments in the 2030 Agenda, indicators 3.7.1, the proportion of women of reproductive age who have their need for family planning satisfied with modern methods of contraception, and 3.7.2, the adolescent birth rate (births per 1,000 women at ages 10–14 and 15–19 years), are used for the global monitoring of progress towards target 3.7, which calls for ensuring, by 2030, universal access to sexual and reproductive health-care services, including for family planning, information and education, and the integration of reproductive health into national strategies and programmes. For indicator 3.7.1, global monitoring relies on updated compilations of data from various sources and the preparation of original estimates of the prevalence of contraceptive use and the demand, or need, for family planning among women of reproductive age, which are published annually by the United Nations [10,11].

The 2030 Agenda calls for the use of data disaggregated along critical social dimensions to identify sub-groups of persons who are being left behind. For family planning indicators, two of the critical dimensions for data disaggregation are woman's age and marital (or union) status. Gaps in knowledge regarding these critical dimensions lead to gaps in service provision, often leaving the most vulnerable and marginalized young women and girls without access to critical reproductive health-care services. Previous research has included comparative analyses based on data from a single survey programme for countries and periods with available data (e.g., the Demographic and Health Survey programme, used in [12–15]) or only provided estimates for adolescents who were married or in a union [16]. As part of the project, *Adding It Up*: *Costs and Benefits of Meeting the Contraceptive Needs of Adolescents*, the Guttmacher Institute prepared estimates of contraceptive use and needs in 2016 among adolescents aged 15–19 living in the developing regions [4] and updated the estimates in 2018 [17]. Additionally, the Guttmacher Institute produced a set of future scenarios of adolescent contraceptive use, cost and impact in developing regions for the period between 2020 and 2030 [18]. Both projects presented estimates for one year or projections according to a specific scenario; results were reported only for specific country groupings based on geographic region or national income per capita.

The purpose of this paper is to respond to the call for improved evidence on adolescent sexual and reproductive health by presenting regional and global estimates and projections of key family planning indicators for adolescent women and girls aged 15 to 19 years in the period from 1990 to 2030. The results are based on a comprehensive compilation of data on contraceptive use by age and marital status (United Nations, 2019b) and methods for deriving model-based estimates and projections of family planning indicators developed by the United Nations [19–21]. These new estimates and projections enable the analysis of levels and trends in contraceptive use among adolescents and of the progress made in meeting their contraceptive needs; they also offer insights about possible future trends in contraceptive use and the need for family planning amongst adolescents between now and 2030, drawing attention to potential areas for accelerated implementation and providing a basis for estimates of both the cost and the impact of increased contraceptive use.

## Data and methods

### Definitions

Women aged 15–49 years are referred to as women of reproductive age, and those aged 15–19 years as adolescent women. For convenience, women who are 'married or in a union' and those who are 'unmarried and not in a union' are referred to as 'married' and 'unmarried', respectively. The report presents results for three key family planning indicators: prevalence of contraceptive use among adolescent women, their unmet need for family planning, and the proportion of their total need for family planning that is being satisfied by modern methods of contraception (see Table 1). The data are presented separately for married, unmarried and all adolescent women.

The major differences in the approaches used for reporting family planning indicators for unmarried women are related to a choice about whether an indicator refers to all women (within a certain age range) or only to those who are deemed to be sexually active. For example, Demographic and Health Survey reports present estimates of contraceptive use and unmet need for family planning for the population of sexually-active unmarried adolescents (as defined by sexual activity in past 28 days), where the denominator includes only sexually-active unmarried adolescents. In this paper, the indicators refer to the population of all unmarried adolescents, and therefore the indicator values also reflect the different extent of sexual activity among adolescents in a given population. Women who are not in need of family planning are women who are not contraceptive users and are not classified as having an unmet need for family planning. For unmarried women, this group includes women who are either sexually inactive (as defined by no sexual intercourse in past 28 days), or sexually active who want to have a child, are infecund or are pregnant or postpartum amenorrhoeic (when the pregnancy was intended). Most unmarried women aged 15–19 years who have no need for family planning are sexually inactive.

The estimates of family planning indicators for married and all adolescents are directly comparable to those generally published in the survey reports (including reports on the Demographic and Health Surveys).

Estimates and projections for individual countries were combined and are presented here for eight geographic regions used for the global monitoring of progress toward the Sustainable Development Goals including 21 geographic subregions and for various country groupings according to the level and history of development. The country classification by income level is based on gross national income per capita, as classified by the World Bank (country groupings are presented as supplementary information, S1 Data).

**Table 1. Definitions of key terms.**

| Term | Definition |
|---|---|
| Contraceptive use | Proportion or number of women who report themselves, or their partners, as currently using at least one contraceptive method of any type. |
| Modern methods of contraception | These include female and male sterilization, the intra-uterine device (IUD), the implant, injectables, oral contraceptive pills, male and female condoms, vaginal barrier methods (including the diaphragm, cervical cap and spermicidal foam, jelly, cream and sponge), the lactational amenorrhea method (LAM), emergency contraception and other modern methods (e.g., the contraceptive patch or vaginal ring). |
| Traditional methods of contraception | These include rhythm (e.g., fertility awareness-based methods, periodic abstinence), withdrawal and other traditional methods. |
| Unmet need for family planning | Proportion or number of women who want to stop or delay childbearing but are not using any contraceptive method. The standard definition of an unmet need for family planning includes women who are fecund and sexually active, but who are not using any method of contraception and i) report not wanting any (more) children, or ii) report wanting to delay the birth of their next child for at least two years or are undecided about the timing of the next birth. It also includes pregnant women whose pregnancies were unwanted or mistimed at the time of conception; and postpartum amenorrheic women experiencing postpartum amenorrhea who are not using family planning and whose last birth was unwanted or mistimed. Infecund women are excluded. The calculation of the indicator follows the 2012 DHS definition. |
| Total need for family planning | Sum of contraceptive use and unmet need for family planning (proportion or number). |
| No need for family planning | Proportion or number of women who are not contraceptive users and are not classified as having an unmet need for family planning. |
| Need for family planning satisfied with modern methods (SDG indicator 3.7.1) | Use of modern contraceptive methods divided by total need for family planning (proportion). |
| Married women | Women who are married or living in a cohabiting union. This category pertains to adolescents who are married (defined in relation to the marriage laws or customs of a country) and to adolescents in a union, which refers to adolescents living with their partner in the same household (also referred to as cohabiting unions, consensual unions, unmarried unions, or "living together"). |
| Unmarried women | Women who are not married and not living in a cohabiting union. |

## Data

**Survey data for key family planning indicators.** Survey-based estimates of contraceptive prevalence by method (any, modern and traditional) and of the unmet need for family planning among adolescent women were compiled in *World Contraceptive Use 2019* [10], a comprehensive collection of family planning indicators, by marital status and by five-year age group, for 195 countries or areas. The observations come mainly from nationally representative household-based surveys and were obtained from the associated micro datasets or derived from tabulations published in reports or specific tabulations provided by the institutions responsible for data collection. For girls and young women aged 15–19 who are married or in a union, the data set includes 754 observations of contraceptive prevalence from 160 countries and 354 observations of unmet need from 101 countries. For adolescent women aged 15–19

who were not married and not in a union, 432 observations from 114 countries on contraceptive prevalence, and 245 observations from 74 countries on unmet need were available for the most recent compilation. The input data set is presented as supplementary information S2 Data. Data for women younger than 15 years of age were not included in this analysis. Data for this age range are rarely collected in surveys or reported separately: the data compilation of *World Contraceptive Use 2019* includes only nine data points from nationally-representative surveys for age groups 10–14 or 12–14.

There are fewer observations for unmarried women, usually because (1) survey questions on current contraceptive use often exclude unmarried women, especially for older surveys, (2) published tables in reports rarely present results for unmarried women, and (3) microdata needed for generating data disaggregated by age and marital status are not available for some surveys [22].

The major sources of family planning data for adolescents used in this study are the Demographic and Health Surveys (DHS) and the Multiple Indicator Cluster Surveys (MICS). Other international survey programmes providing at least 25 observations are the Reproductive and Health Surveys, the World Fertility Survey, and the Performance Monitoring and Accountability Surveys. The proportions of observations obtained from these sources are given in Table 2. One consequence of relying on data from these survey programmes is that geographic coverage is best in their focus regions, namely sub-Saharan Africa, Central and Southern Asia, Western Asia and Northern Africa, Eastern and South-Eastern Asia, and Latin America and the Caribbean (data availability by regions are presented as supplementary information, Figure A in S1 Supplementary Figures in S1 File). There are only a few observations of the unmet need for family planning among adolescents in Northern America and Europe, Oceania, and Eastern and South-Eastern Asia.

The survey data on contraceptive use among adolescents cover the period 1969 to 2018, with the majority of surveys (84% for married, 94% for unmarried) conducted after 1990. Furthermore, 57% of countries had no data from before 1990 for married adolescents, and 80% had no data before 1990 for unmarried adolescents. At the regional level, 47 countries in sub-Saharan Africa had at least one observation of contraceptive prevalence for married adolescents; however, before 1990 70% of these countries had no observations and a further 21% had only one. For unmarried adolescents, 42 sub-Saharan African countries had at least one observation; of these, 76% had no observations before 1990. Countries in Latin American and the Caribbean had more early observations, 82% of countries in this region had a least one observation before 1990 for married women, and 42% for unmarried women. Additionally, the number of survey observations varies by country. Of the 160 countries with at least one observation for married women, 41% had five or more observations, and 30% of the 114 countries with at least one observation for unmarried women had five or more observations.

**Estimates and projections of the proportion and number of adolescents aged 15–19 years who are married or in a union.** The estimated population of adolescent women by marital status and country or area were used to transform model-based estimates of proportions into estimates of counts of women using family planning. These population counts were

**Table 2. Distribution of observations by survey type, for married and unmarried adolescent women aged 15–19 (in per cent).**

| | Demographic and Health Surveys (DHS) | Multiple Indicator Cluster Surveys (MICS) | Reproductive Health Surveys (RHS) | World Fertility Surveys (WFS) | Performance Monitoring and Evaluation Surveys (PMA) | Other Surveys | Total | Number of surveys |
|---|---|---|---|---|---|---|---|---|
| **Married** | 42 | 18 | 6 | 4 | 5 | 25 | 100 | 754 |
| **Unmarried** | 54 | 21 | 2 | 0 | 8 | 15 | 100 | 432 |

**Table 3. Proportion of women aged 15–19 years old who are married or in a union, globally and by region, 1990–2030.**

| Region | 1990 | 2000 | 2010 | 2019 | 2030 |
|---|---|---|---|---|---|
| World | 18.5 | 17.1 | 14.2 | 12.3 | 9.3 |
| Sub-Saharan Africa | 32.1 | 28.1 | 24.2 | 20.5 | 14.9 |
| Western Asia and Northern Africa | 17.9 | 14.5 | 11.5 | 10.2 | 8.0 |
| Central Asia and Southern Asia | 37.5 | 29.3 | 21.1 | 14.8 | 10.0 |
| Eastern and South-eastern Asia | 7.2 | 6.9 | 6.0 | 5.6 | 4.9 |
| Latin America and the Caribbean | 15.3 | 15.0 | 15.0 | 14.4 | 11.7 |
| Oceania | 7.8 | 7.8 | 7.2 | 6.8 | 5.7 |
| Northern America and Europe | 8.1 | 6.6 | 4.8 | 4.0 | 3.7 |

derived from the 2018 revision of *Estimates and Projections of Women of Reproductive Age Who Are Married or in a Union* [23], including estimates and projections of the proportion of women who are married or in a union disaggregated by five-year age group for 201 countries or areas over the period 1970 to 2030. Estimates are based on selected census- or survey-based observations of the proportion of female population by marital status and age from *World Marriage Data* 2017 [24].

Changes in the proportions of adolescents who are married or in a union influence the proportions of contraceptive users and those with unmet need for family planning among all adolescents since contraceptive use and needs vary considerably by marital status. The proportions of adolescents aged 15–19 that are married or in a union in the regions are given in Table 3 and Fig 1. The proportions married have declined in all regions and are projected to continue declining to 2030 [23]. Worldwide, 18.5% of women in this age group were married or in a union in 1990, but this was estimated to be 12.3% in 2019 and is projected to be 9.3% in 2030. Central Asia and Southern Asia, Sub-Saharan Africa, and Western Asia and Northern Africa had the greatest proportions married in 1990 (37.5%, 32.1% and 17.9%, respectively) and are projected to continue fast declines through 2030. Conversely, Eastern and South-Eastern Asia, Oceania, and Northern America and Europe had low, and still declining, proportions married or in a union. Latin America and the Caribbean has seen only a little change in the proportion married among adolescents aged 15–19, from 15.3% in 1990 to 14.4% in 2019; this proportion is projected to decline only moderately to 11.7% in 2030, which will be the highest level for any region other than sub-Saharan Africa.

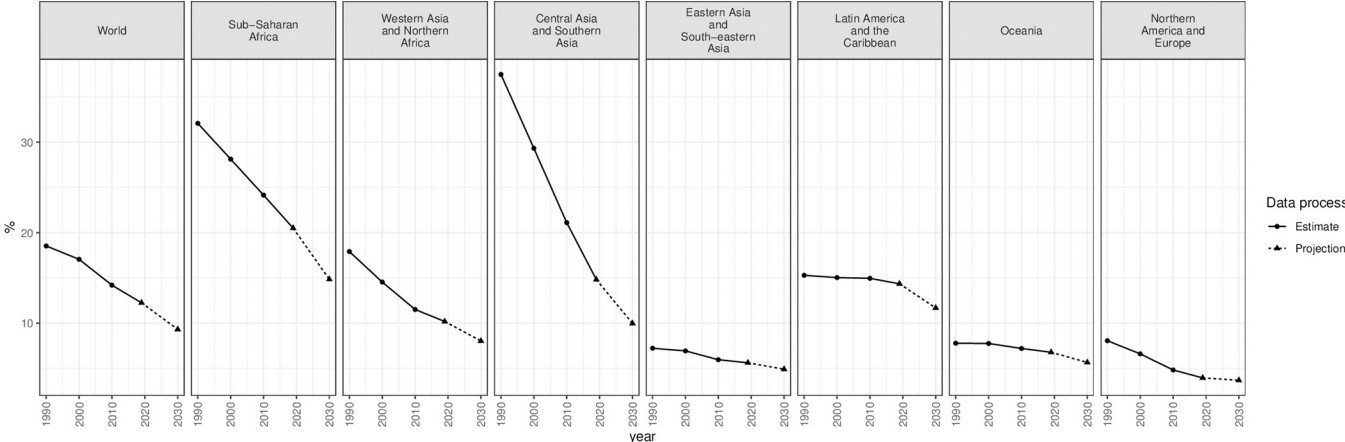

**Fig 1. Proportion of women aged 15–19 years who are married or in a union, for the world and SDG level 1 regions, 1990–2030.**

Some trends in proportions married by regions were due primarily to the dynamics in sub-regions. In particular, the decline in Central and Southern Asia was driven primarily by the trend in Southern Asia. Results by subregion, development group and income group are presented as S2 Data.

## Statistical methods

**Modelling trends in contraceptive use and unmet need for family planning.** A Bayesian hierarchical model was used to estimate and project contraceptive prevalence and the unmet need for family planning among women aged 15–19 years. The estimates for women who are married or in a union were derived separately from those for women who are not married or in a union. We used the same model for estimating and projecting family planning indicators developed in earlier research for all women of reproductive age (15–49 years) [19–21], except we applied it to data only on women aged 15–19 years. We refer the reader to the supplementary material of [21] for the full model specification; we give a conceptual overview below.

For each country, the model generates estimates of the proportion of adolescents aged 15–19 belonging to each of the following four categories: i) using a traditional contraceptive method, ii) using a modern contraceptive method, iii) having an unmet need for family planning, and iv) having no need for family planning. The shares across the four groups must sum to one. In the modelling process [19], an equivalent set of quantities is modelled: i) the proportion using a contraceptive method of any kind, ii) among those using any method of contraception, the proportion using any modern method, iii) among those not using any method of contraception, the proportion who have an unmet need. From any three of these indicators, the fourth can be calculated, bearing in mind that the four indicators must sum to one.

The time trends for the proportion of women using contraception (any method) and the proportion using modern contraceptive methods were modelled using logistic curves. These are "S"-shaped curves that capture the entire transition from a state of very low contraceptive prevalence, through a period of growth, to an eventual maximum level. A separate pair of curves was estimated for each country. Thus, for example, the rate of growth in prevalence during the transition period, the timing, and the long-run maximum level could, and did, vary by country. Moreover, the model did not assume, nor impose the condition, that all countries move through the entire curve during the period for which estimates and projections were prepared. Many countries, for example, remained in the state of low prevalence across the whole time period of interest, including to the end of the projection. In these cases, only a portion of the logistic curve was used to model the components. The extent of the curve used was determined by the country-level data through the statistical model.

The proportion experiencing unmet need for family planning, the third of the modelled parameters, was not modelled as a logistic curve over time but as a function of the total contraceptive prevalence. This was based on theoretical reasoning about the relationship between these two indicators. The reasoning was that, as prevalence increases, there will be a concomitant increase in unmet need due to an increased need for, and awareness of, contraception as it becomes more common in the population. At a certain level of contraceptive prevalence, unmet need reaches a turning point and begins to decline as availability, accessibility, and acceptability of contraception improves and use increases further. The model used encapsulates this relationship.

**Accounting for bias due to sampling of non-baseline groups and misclassifications.** Some surveys targeted groups outside the baseline national populations of women aged 15–19. Examples include the Gender and Generations Surveys (GGSs) that surveyed only women aged 18 and over. Several other surveys excluded some populations, for example non-nationals

or people living in particular provinces or areas. Additional sources of bias were due to mis-classification of women as users of modern or traditional methods. These sources of bias were accounted through the use of additional parameters, multipliers and misclassification parameters, that adjusted the estimated proportions to counter the sources of bias. These were estimated from the data through the statistical model in the same way as all the other model parameters. A detailed explanation of these parameters is given in previous work [21]. The input data file, including biases and misclassifications, is presented as supplementary information S3 Data.

**Hierarchical structures of models for married and unmarried women.** Many countries had few data points and few countries had no data available. To improve the precision and accuracy of estimations and projections, hierarchical models were used [25]. These models allowed for 'borrowing of strength' across data points such that estimates for countries with little-to-no data were based partly on data for other 'similar' countries. What constituted 'similar' was defined through the hierarchical structure of the model. For married women, a geographic hierarchy wherein countries were clustered into sub-regions (e.g., Eastern Africa), and sub-regions into regions (e.g., Africa) was used [19]. All countries belonged to one, and only one, sub-region and region. All regions together constituted the world. The clustering was based on geographic location, regardless of the availability of data. For further details on how hierarchies are used in the model, see S1 Appendix.

For unmarried adolescent women, need for family planning is closely related to level of sexual activity and sexual activity among unmarried adolescent women varies considerably between countries [26]. To account for this association, a different hierarchical structure was used for unmarried women. Countries were first clustered into geographic sub-regions similar to those used in the model for married women. The sub-regions were then clustered into one of two sexual activity groups [20]. Group 0 consisted of countries where sexual activity among unmarried women was estimated to be very low, defined as the proportion of unmarried women of reproductive age who report sexual activity in past 28 days was not more than 2%. Group 1 consisted of all other countries. To estimate the extent of sexual activity among unmarried women was estimated from the question on recent sexual activity in DHS, MICS and other available surveys. In countries without this data, we used information about the acceptance of sex between unmarried adults from other surveys and the level of religiosity. For further details on the information used for creating the clusters, see S1 Appendix.

**Projections to 2030.** Estimates for years outside the periods of data availability were obtained from the fitted Bayesian model. The parameters of the systematic trend component are time-invariant [19, Technical Appendix] and thus provide estimates of the trend at all years. The auto-correlated error process is also parameterized by time-invariant parameters. Once these were obtained from the model fit, extrapolations beyond the period of data availability were obtained by sequentially sampling from the error process conditional distributions.

**Parameter estimation and software.** A Markov chain Monte Carlo (MCMC) algorithm [25,27] was used to draw a large sample from the posterior joint probability distribution of the parameters described above (i.e., the proportion of women using any method of contraception, the proportion of users using modern methods, the proportion of non-users experiencing unmet need), by country, by year. These were transformed into country-level estimates of the indicators of interest, such as contraceptive prevalence of modern methods or unmet need for family planning.

The MCMC samples were used to generate summary statistics for any parameter of interest. For example, the posterior median contraceptive prevalence, which was used as the measure of central tendency, was obtained by taking the empirical median of the marginal MCMC sample

for that parameter. We followed Kantorová et al. [22, S1 Appendix, Subsection 3.7.6] and used small adjustments to medians only to ensure that basic identities held, such as that the proportion using any method equal the sum of the proportions using modern and traditional methods.

Out-of-sample validation exercises were performed to assesses model fit; these are reported in S1 Appendix. The exercises indicated that the model fitted the data well and had good out-of-sample predictive validity.

All computation was done using the *R Environment for Statistical Computing*, version 3.5.1 (R Core Team, 2018) and *JAGS* version 4.2.0 [28,29]. The source code is available at https://github.com/FPcounts/FPEMglobal.

**Indicators reported.** Estimates and projections of family planning indicators among adolescent women aged 15–19 are presented for: contraceptive prevalence (any method, modern methods), the unmet need for family planning, the proportion of need for family planning satisfied by modern contraceptive methods (SDG indicator 3.7.1), and total need for family planning. These are given for the groupings by regions and subregions (the SDG country groupings at levels 1 and 2, supplementary information S1 and S2 Results), income levels and development groups (supplementary information S3 and S4 Results) as the proportions and the counts. For the proportion of need for family planning satisfied by modern contraceptive methods (SDG indicator 3.7.1.), the results for the regions of "Western Asia and Northern Africa" and "Central Asia and Southern Asia" are not presented for unmarried adolescent women separately due to small overall counts of women with need for family planning; however, they are included in the all adolescent estimates.

The results are presented for individual countries in supplementary information S2 Appendix. Estimates and projections for countries with fewer than two survey-based observations of the proportion of married adolescents aged 15–19 using any method of contraception are not presented individually because they are subject to substantial uncertainty. However, they are included in the aggregates.

The 95% uncertainty intervals (UI) are computed for all indicators using the empirical 2.5th and 97.5th percentiles of the marginal posterior distributions and reported in tables and figures. Empirical medians (50th percentiles) of the marginal posterior distributions were used as single-number summaries of central tendency.

As an example, modelled estimates for Colombia are shown with the underlying source data in Fig 2. The modelled estimates do not necessarily pass through all data points because they account for systematic bias and uncertainty due to measurement error.

In many countries, the need for family planning and contraceptive use are substantially different among married and unmarried adolescents. Therefore, we present the results by marital status, as well as for all adolescents.

Changes over time in the proportion of adolescents who are married or in a union can have a strong effect on overall need for family planning and contraceptive use. Estimates for all women were derived for each indicator by weighting and summing the respective estimates for married and unmarried women. The proportions of adolescent women who were married or unmarried were used as weights. Summing was done at the level of the MCMC samples by country. This yielded a joint posterior distribution for all women which was summarized using quantiles in the same way as described above for married and unmarried women.

Estimates for geographical aggregates were derived by summing MCMC samples at the country level to produce joint posterior distributions.

For each indicator, estimates of the proportion of all women of reproductive age that are adolescents were calculated by taking the ratio of the number of adolescent women in each indicator category to the number of all women of reproductive age [11].

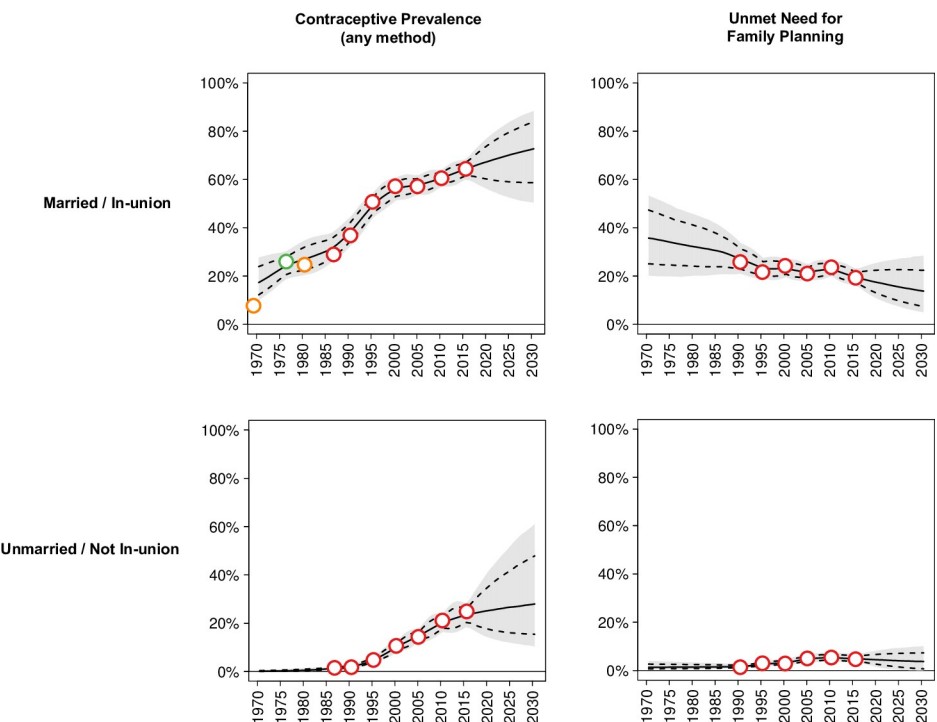

**Fig 2. Estimates and projections of contraceptive prevalence (any method) (left panels) and unmet need for family planning (right panels), and source data, among married (top panels) and unmarried (bottom panels) women aged 15–19 years in Colombia, 1970–2030.** Note: Estimates and projections are indicated by the solid line (posterior medians), dashed lines (lower and upper limits of 80% uncertainty/prediction interval), and grey ribbon (extent of 95% uncertainty/prediction interval). The source data are indicated by the red (DHS), green (other international surveys) and orange (national surveys) circles.

**Joint visualization of three family planning indicators by country.** In addition to reporting trends, visualizations are used to display family planning indicators jointly on maps. This was possible because the three indicators form a complete partition of the population, with each woman being counted under one, and only one, of the indicators. For this reason, at the population level the indicator values sum to 100%. In general, indicators with this property are called compositions [30], as illustrated in Fig 3.

Compositional indicators were formed from the posterior medians for each marital group, for the year 2019. These were displayed as choropleth world maps with ternary plots and ternary-balance colour schemes [31]. These visualizations show the distribution of countries among the three components in true multivariate fashion and illustrate the relationships among the three component indicators simultaneously across countries.

The ternary balance colour scheme assigns a basic colour to each of the three components and represents compositions by mixtures of these colours in the same proportions. This gives each composition a single, unique colour. We assigned green to contraceptive prevalence, pink to unmet need, and blue to no need. In the extreme, countries with a proportion of no need close to 100% would be coloured almost purely blue. Countries with proportions more evenly allocated would be coloured an appropriate blend of the base colours, where the proportion of each base colour is equal to the proportion of each component. For example, a country with the proportions of women using contraception of 65%, having an unmet need for family planning of 25%, and having a no need for family planning of 10%, would be indicated on the map by the colour obtained by blending green, pink, and blue in relative proportions of 65%, 25%, and 10%.

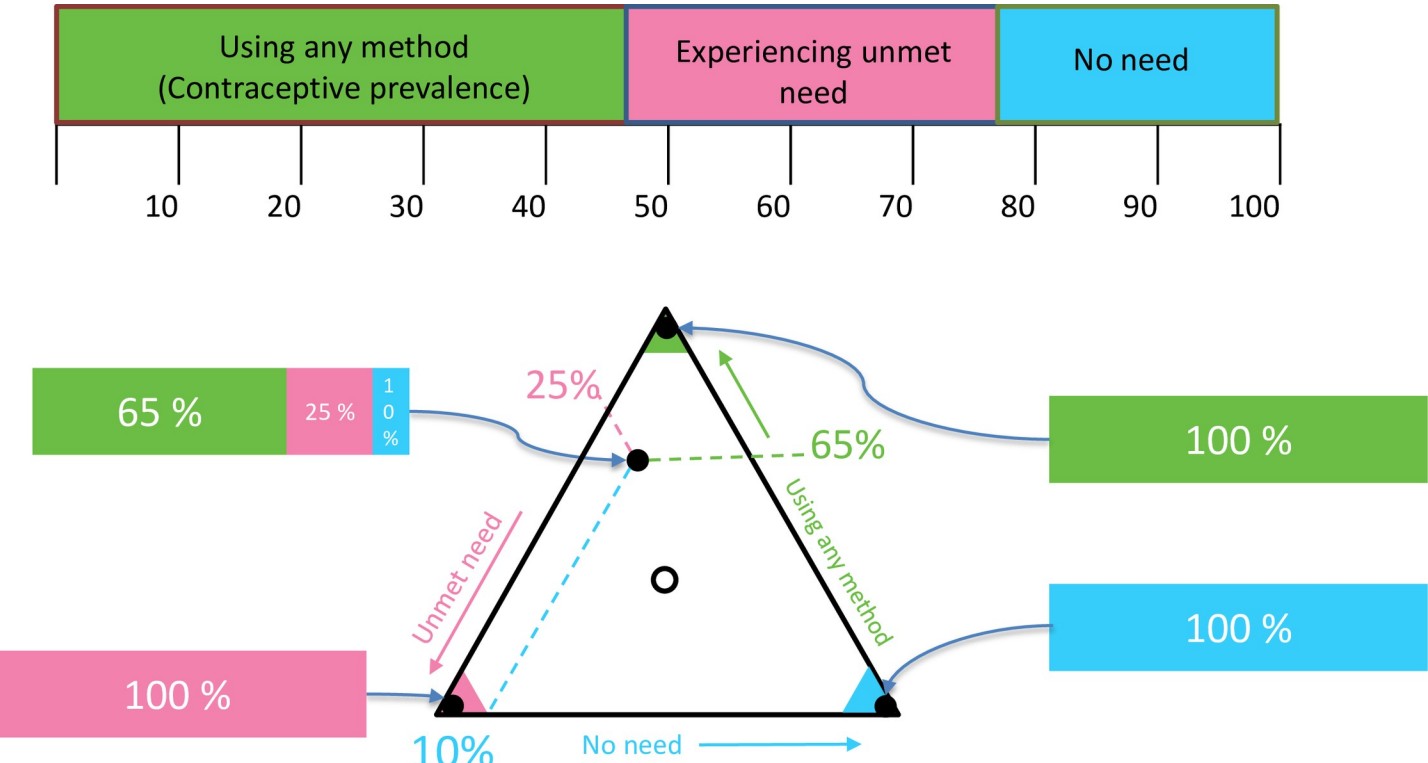

**Fig 3. Family planning indicators as categories of a composition (top panel) and illustration of family planning compositions plotted on a ternary diagram (bottom panel).** Note: The bars represent compositions; vectors of components that sum to 100%. The points inside the triangle plot the values for specific hypothetical countries. The right edge is the axis for contraceptive prevalence, the left edge for unmet need, and the bottom edge is the axis for no need. All axes range from 0 to 100%. The arrows adjacent to the triangle's edges indicate the direction of increasing values; they point to the vertex at which that component is 100%. The open circle marks the centre of the triangle; at this point all components are equal at $33^1/_3$%.

The legend for the ternary balance choropleth maps is the ternary diagram overlaying the ternary colour scheme. A ternary diagram is a triangular plotting space, resulting in a legend that is a triangle with a colour contour inside it. Each edge of the triangle is an axis for one of the components. The interior regions of the triangle near the vertices are made up entirely of the base colours corresponding to the component that is at its maximum at that vertex. In the deep interior of the triangle, the colours become a blend as described above.

The posterior median estimates for each country are plotted as points inside this triangular legend just as they are in the rectangular plotting space of a standard scatter plot. Points close to the vertices represent countries where most women fall into the component associated with that vertex. Points near the centre have proportions for each component close to 1/3. Values of the three components for any point are found by reading back to each edge in the directions indicated by the orientations of the axis labels, as illustrated in Fig 3.

## Results

### Need for family planning

Worldwide, 15.4% (95%UI = 13.4–21.8) of adolescent women aged 15–19 are estimated to have a need for family planning (Table 4). The differences by region and marital status, and the changes over time, are important (see Supplementary information S2 and S4 Results).

Among married adolescent women, the total need for family planning increased between 1990 and 2019 from 44% (95%UI = 40.8–47.5) to 51.6% (95%UI = 49.1–54.3). Among

**Table 4. Estimates and projections of the proportions of adolescent women (15–19 years) using contraception (any method and modern methods) and having an unmet need for family planning, by marital status and region, in 2019, in per cent.**

| Marital status | Region | Contraceptive use (any method) Median (95% UI) | Contraceptive use (modern methods) Median (95% UI) | Unmet need for family planning Median (95% UI) | Need for family planning Median (95% UI) | Proportion of need for family planning satisfied by modern methods Median (95% UI) |
|---|---|---|---|---|---|---|
| All women | World | 10.2 (8.4, 14.3) | 9.1 (7.4, 12.8) | 5.1 (4.2, 10.0) | 15.4 (13.4, 21.8) | 59.2 (44.8, 67.2) |
| | Sub-Saharan Africa | 11.1 (10.0, 12.5) | 9.7 (8.8, 10.9) | 10.2 (9.2, 11.5) | 21.4 (19.8, 23.3) | 45.6 (42.2, 49.0) |
| | Western Asia and Northern Africa | 3.4 (2.5, 12.7) | 2.4 (1.8, 8.4) | 2.8 (1.9, 7.6) | 6.2 (4.8, 16.4) | 38.7 (20.9, 56.5) |
| | Central and Southern Asia | 4.1 (3.4, 7.5) | 3.1 (2.6, 5.3) | 3.1 (2.5, 3.9) | 7.2 (6.4, 10.7) | 43.5 (36.6, 52.3) |
| | Eastern and South-Eastern Asia | 6.1 (3.2, 21.8) | 5.4 (2.8, 20.1) | 2.5 (1.0, 22.4) | 8.6 (4.9, 34.9) | 62.3 (19.4, 85.1) |
| | Latin America and the Caribbean | 25.3 (18.6, 35.8) | 23.7 (17.2, 34.0) | 7.2 (4.6, 12.7) | 32.4 (25.7, 42.8) | 73.1 (60.0, 83.3) |
| | Oceania | 19.2 (6.3, 37.4) | 18.2 (5.4, 36.3) | 4.8 (1.5, 30.9) | 24.0 (10.1, 51.5) | 76.0 (27.5, 92.0) |
| | Northern America and Europe | 23.5 (15.4, 32.9) | 22.2 (14.3, 31.4) | 3.9 (1.2, 20.4) | 27.4 (19.0, 44.1) | 81.1 (47.5, 91.6) |
| Married | World | 30.9 (28.7, 33.3) | 26.2 (24.2, 28.4) | 20.7 (18.7, 23.0) | 51.6 (49.1, 54.3) | 50.8 (47.3, 54.2) |
| | Sub-Saharan Africa | 17.9 (16.1, 19.8) | 15.4 (13.7, 17.2) | 23.7 (21.4, 26.4) | 41.6 (38.9, 44.6) | 37.0 (33.4, 40.7) |
| | Western Asia and Northern Africa | 24.0 (19.5, 29.4) | 17.4 (13.9, 21.7) | 18.1 (14.0, 23.0) | 42.1 (36.5, 48.4) | 41.2 (33.9, 48.8) |
| | Central Asia and Southern Asia | 25.1 (21.0, 29.9) | 19.8 (16.2, 23.9) | 20.9 (16.7, 26.0) | 46.0 (40.8, 52.1) | 43.0 (35.7, 50.4) |
| | Eastern Asia and South-eastern Asia | 49.0 (39.7, 58.8) | 43.9 (35.2, 53.5) | 16.8 (11.8, 23.9) | 65.8 (57.5, 73.8) | 66.7 (56.5, 75.8) |
| | Latin America and the Caribbean | 63.0 (55.2, 70.0) | 57.3 (49.7, 64.2) | 18.4 (14.0, 23.8) | 81.4 (76.4, 85.7) | 70.3 (63.1, 76.4) |
| | Oceania | 35.8 (24.4, 51.3) | 28.7 (16.3, 42.6) | 21.2 (10.7, 36.5) | 57.0 (43.3, 73.9) | 50.4 (29.5, 68.4) |
| | Northern America and Europe | 71.4 (63.6, 78.2) | 61.2 (51.7, 69.3) | 11.2 (6.9, 17.4) | 82.7 (76.1, 88.6) | 74.0 (63.8, 81.6) |
| Unmarried | World | 7.4 (5.3, 12.0) | 6.7 (4.9, 10.9) | 3.0 (1.9, 8.5) | 10.3 (8.1, 17.6) | 65.1 (42.0, 76.3) |
| | Sub-Saharan Africa | 9.4 (8.2, 11.0) | 8.3 (7.2, 9.6) | 6.7 (5.8, 8.0) | 16.1 (14.4, 18.2) | 51.3 (46.6, 55.9) |
| | Western Asia and Northern Africa | — | — | — | — | — |
| | Central Asia and Southern Asia | — | — | — | — | — |
| | Eastern Asia and South-eastern Asia | 3.6 (0.6, 20.3) | 3.1 (0.5, 18.6) | 1.7 (0.2, 22.7) | 5.2 (1.3, 33.0) | 59.0 (8.2, 91.6) |

*(Continued)*

**Table 4.** (Continued)

| Marital status | Region | Contraceptive use (any method) Median (95% UI) | Contraceptive use (modern methods) Median (95% UI) | Unmet need for family planning Median (95% UI) | Need for family planning Median (95% UI) | Proportion of need for family planning satisfied by modern methods Median (95% UI) |
|---|---|---|---|---|---|---|
| | Latin America and the Caribbean | 18.9 (11.4, 31.1) | 18.1 (10.8, 30.0) | 5.3 (2.5, 11.6) | 24.2 (16.5, 36.2) | 74.7 (55.2, 88.4) |
| | Oceania | 17.9 (4.2, 37.3) | 17.4 (3.8, 36.7) | 3.6 (0.3, 31.6) | 21.5 (6.7, 51.1) | 80.9 (24.2, 97.1) |
| | Northern America and Europe | 21.5 (13.1, 31.3) | 20.6 (12.5, 30.1) | 3.6 (0.8, 20.8) | 25.1 (16.4, 42.4) | 82.0 (45.3, 93.7) |

Note: Results are 'median (lower, upper)', where 'median' is the posterior median and 'lower' and 'upper' are the lower and upper limits of the posterior 95% uncertainty intervals.

unmarried adolescent women, it was low in 1990, at 6.2% (95%UI = 3.9–18). Even though the median estimate had increased to 10.3% (95%UI = 8.1–17.6) in 2019, estimation uncertainty is too high to conclude that there was in fact an upward trend. Similarly, median projections indicate an increase among both married and unmarried adolescent women between 2019 and 2030. Again, however, the uncertainty surrounding these forecasts is substantial: the projected level in 2030 is 56.7% (95%UI = 52.4–61.4) for married adolescent women and 11.9% (95% UI = 9.1–19.3) for unmarried adolescent women.

By region, the total need for family planning in 2019 among unmarried adolescent women was highest in Northern America and Europe, Latin America and the Caribbean, and Oceania; in sub-Saharan Africa it was above the global average at 16.1% (95%UI = 14.4, 18.2). The higher needs for family planning in these regions are related to higher sexual activity among unmarried adolescents compared to other regions. The total need for family planning among married adolescent women in sub-Saharan Africa in 2019 was the lowest among all regions, at 38.9% (95%UI = 37.4–41.6), and it was below 50% also in Central and Southern Asia, and Western Asia and Northern Africa. In these regions, among adolescents who are married or in union, the pregnancies are to a large degree intended. In other regions, the need for family planning among married adolescents is high and has increased since 1990.

## Contraceptive use

Worldwide, 10.2% (95%UI = 8.4–14.3) of adolescent women aged 15–19 years were using some method of contraception in 2019, with the highest proportions of users among adolescents in Latin America and the Caribbean, at 25.3% (95%UI = 18.6–35.8), and in Northern America and Europe, at 23.5% (95%UI = 15.4–32.9) (Table 4).

There are large differences in contraceptive use among married and unmarried adolescents. The proportion of unmarried adolescents using any contraceptive method increased from 3.8% (95%UI = 2.2–8.6) in 1990 to 7.4% (95%UI = 5.3–12.0) in 2019 (Fig 4 and Table 4). It is projected to reach 8.6% (95%UI = 6.0–14.3) in 2030. The proportion of married adolescents using a contraceptive method was higher throughout the observation period, increasing from 15.2% (95%UI = 13.9–16.9) in 1990 to 30.9% (95%UI = 28.7–33.3) in 2019, with a projection of 37.2% (95%UI = 32.5–42.6) in 2030.

The proportion of contraceptive users among both married and unmarried adolescents in 2019 was highest in Northern America and Europe, at 71.4% (95%UI = 63.6–78.2) for married and 21.5% (95%UI = 13.1–31.3) for unmarried women, and in Latin America and the Caribbean,

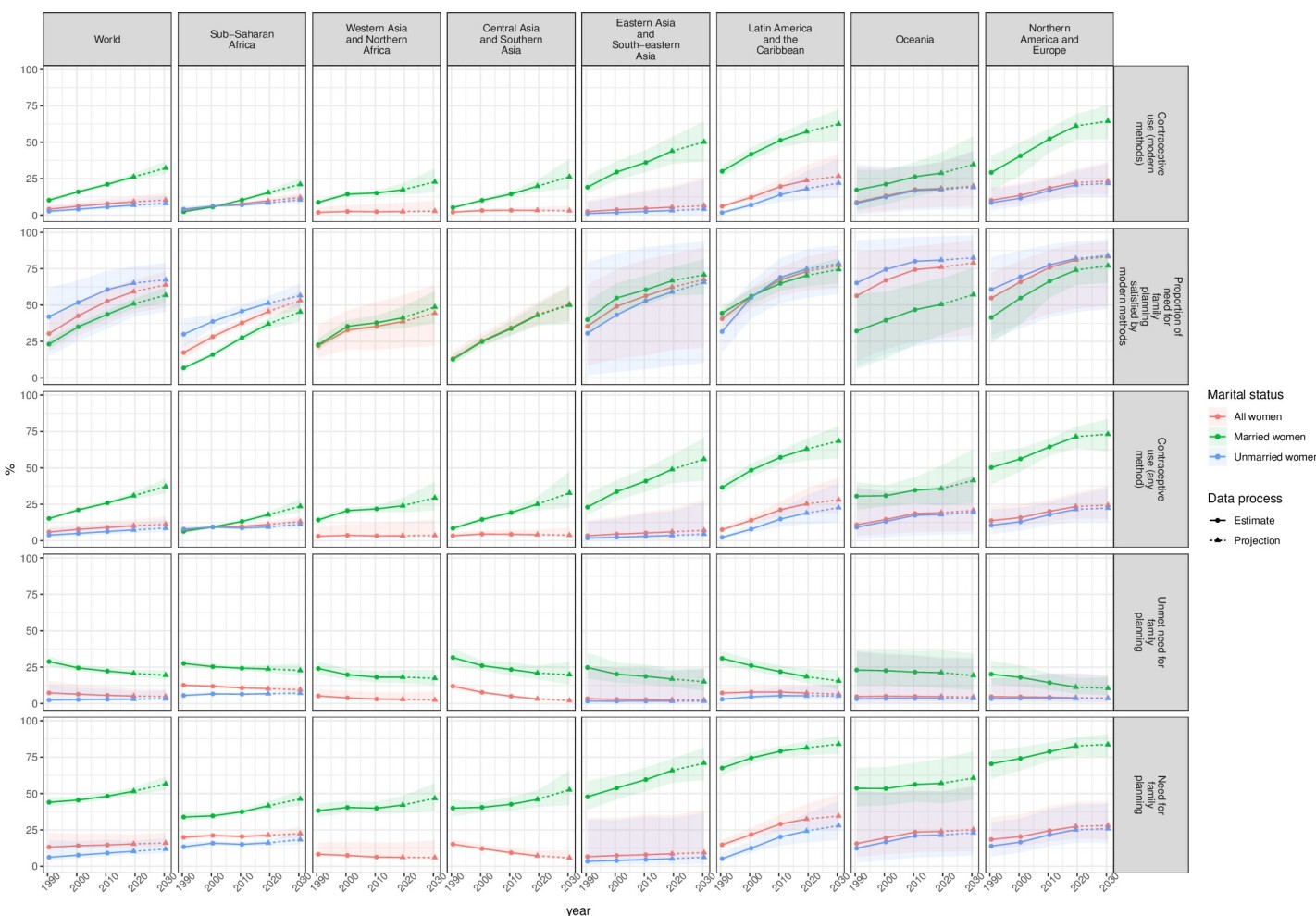

**Fig 4. Estimates and projections of the proportion of adolescent women (15–19 years) using contraception (any method and modern methods) and having unmet need for family planning, by marital status and regions, 1990–2030.** Given are the posterior medians (solid lines) posterior 95% uncertainty intervals (ribbons).

at 63.0% (95%UI = 55.2–70.0) for married and 18.9% (95%UI = 11.4–31.1) for unmarried women. Eastern and South-Eastern Asia had the third highest prevalence in 2019 among married adolescents, at 49.0% (95%UI = 39.7–58.8), but had a low proportion of users among unmarried adolescents, at 3.6% (95%UI = 0.6–20.3), reflecting relatively low levels of sexual activity among unmarried adolescents compared to those living in other regions. In sub-Saharan Africa in 2019, the proportion of contraceptive users among married adolescents was the lowest among all regions at 17.9% (95%UI = 16.1–19.8), but for unmarried adolescents, the region had the fourth highest level of contraceptive use, at 9.4% (95%UI = 8.2–11.0), reflecting a higher average level of sexual activity among unmarried women compared to those in other regions (Ueffing et al, 2019). In 2019, the proportions of contraceptive users among unmarried adolescents in Central and Southern Asia, and in Western Asia and Northern Africa, were the lowest among the regional groupings considered here. In these two regions, the estimated level of sexual activity among unmarried women in general, and among unmarried adolescents in particular, is low, accounting for the low levels of need for family planning and use of contraception.

Since 1990, the proportion of married adolescents using contraception increased in all regions, and the proportion of unmarried adolescents using contraception increased most in Northern America and Europe, and Latin America and the Caribbean.

**Contraceptive use (modern methods).** Globally in 2019, the estimated proportion of adolescent women (ages 15–19 years) who were using modern methods of contraception was 9.1% (UI = 7.4–12.8) (Table 4). Majority of adolescent users report using a modern method, 89.2% (95%UI = 83.0–91.9) in 2019 increasing from 68.5% (95%UI = 54.7–77.5) in 1990. The proportion of modern contraceptive use among all use among adolescents was highest in Northern America and Europe, at 94.6% (95%UI = 83.0–91.9) and it was lowest in Western Asia and Northern Africa, at 71.0% (95%UI = 59.5–79.8).

**Unmet need for family planning.** Worldwide, 5.1% (95%UI = 4.2–10.0) of adolescent women aged 15–19 years are estimated to have an unmet need for family planning, with the highest proportions of unmet need among adolescents in sub-Saharan Africa at 10.2% (95% UI = 9.2–11.5) (Table 4 and Fig 4).

The proportion of married adolescents with unmet need for family planning worldwide in 2019 was estimated to be 20.7% (95%UI = 18.7–23.0) and among unmarried adolescents 3.0% (95%UI = 1.9–8.5). In view of the wide prediction intervals, the projections to 2030 are inconclusive concerning likely future changes in the proportion of adolescents with an unmet need for family planning at the global level: the projected level of this indicator in 2030 is 19.5% (95%UI = 16.6, 22.8) for married adolescent women and 3.3% (95%UI = 2.1–8.7) for unmarried adolescent women.

The proportion of adolescents with an unmet need for family planning in 2019 among married adolescent women was highest in sub-Saharan Africa at 23.7% (95%UI = 21.4–26.4). Among unmarried adolescent women in the same year, the regions with the greatest unmet need for family planning were sub-Saharan Africa at 6.7% (95%UI = 5.8–8.0), and Latin America and the Caribbean at 5.3% (95%UI = 2.5–11.6).

## Need for family planning satisfied by modern methods

The proportion of adolescent women and girls aged 15–19 who had their need for family planning satisfied by modern methods (SDG indicator 3.7.1.) was 59.2% (95%UI = 44.8–67.2) globally in 2019 (Table 4 and Fig 5). Among married adolescents, it was 50.8% (95%UI = 47.3–54.2); among unmarried adolescents it was 65.1% (95%UI = 42–76.3). Across all regions, the proportion of need satisfied by modern methods was highest in all marital groups in Northern America and Europe. In contrast, it was less than one half in Western Asia and Northern Africa, at 38.7% (95%UI = 20.9–56.5), in Central and Southern Asia, at 43.5% (95%UI = 36.6–52.3), and in sub-Saharan Africa, at 45.6% (95%UI = 42.2–49.0).

Among all women of reproductive age (15–49 years), the proportion of the need for family planning satisfied by modern methods, Sustainable Development Goals (SDG) indicator 3.7.1, was 75.7% (95% UI 73.2%–78.0%) globally [22]. In nearly all countries, median estimates of the proportion of the need for family planning satisfied by modern methods is lower among 15–19 years old compared to 15–49 years old (Fig 6).

## Joint visualization of three family planning indicators by country

The compositions of adolescent women populations in terms of contraceptive prevalence (any method), unmet need, and no need for family planning in 2019 are illustrated in the ternary colour scheme maps in Fig 7 (married), Fig 8 (unmarried), and Fig 9 (all women).

Amongst married adolescents, there is considerable variation across the three components. Married (or in cohabiting union) adolescents in Western Europe are characterised by having a high prevalence of contraceptive use, relatively low levels of unmet need and low levels of no need (light green in the map). Married adolescents in countries depicted in dark green (Eastern Europe, Eastern and South-eastern Asia, Zimbabwe) are those with a moderate prevalence

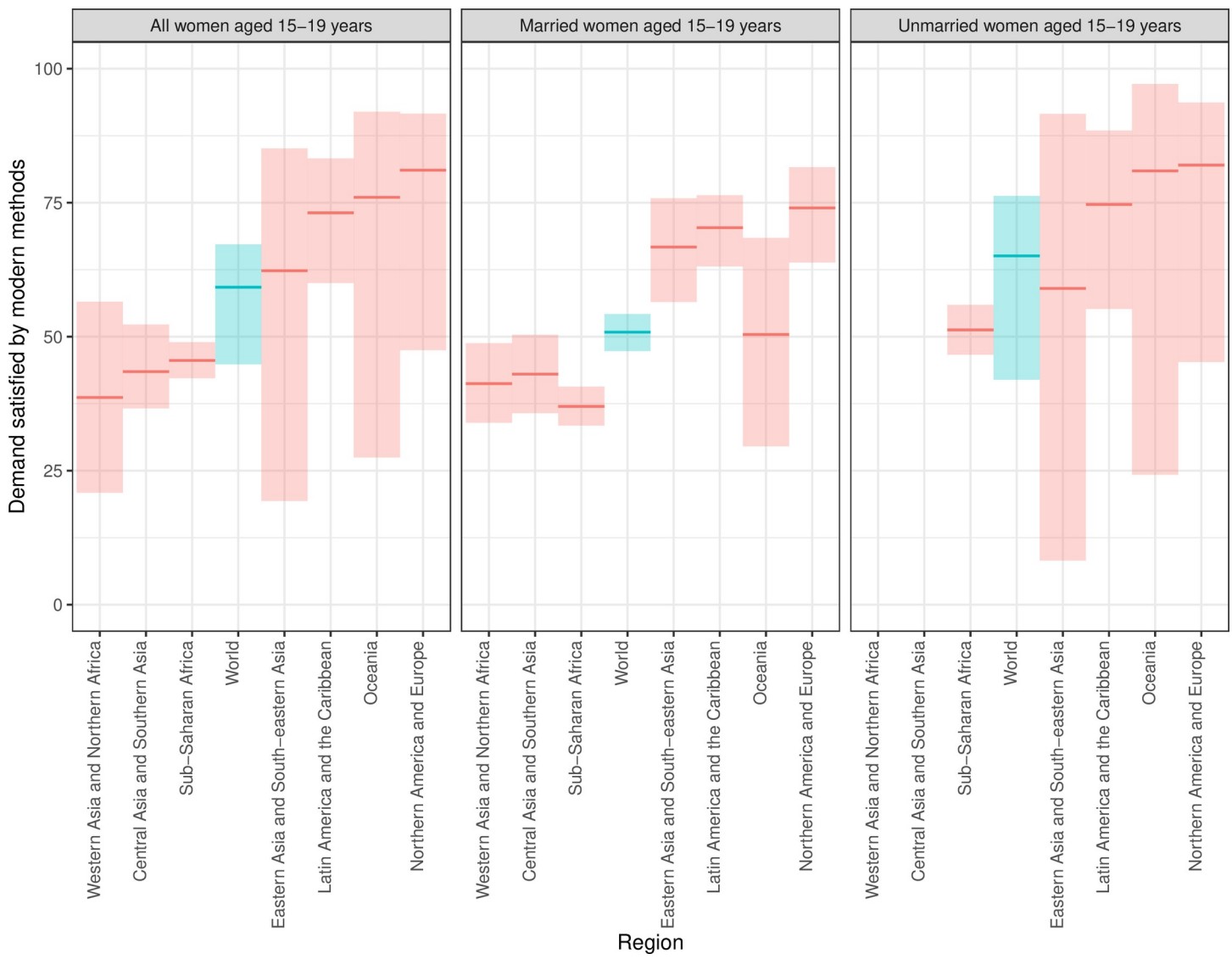

**Fig 5. Proportion of need for family planning satisfied by modern methods among adolescent women (ages 15–19 years) in 2019, by region.** Note: Ribbons indicate the extent of posterior 95% uncertainty intervals, solid lines indicate posterior medians. To facilitate comparisons, regions in are plotted, left to right, in order of increasing value among all adolescent women.

of contraception use (40–60%), moderate levels of no need (30–50%), and substantial levels of unmet need (10–20%). Married adolescents in parts of Middle and Western Africa (purple) have high levels of unmet need, a low prevalence of contraceptive use, and moderate levels of no need. Amongst married adolescents in India and much of sub-Saharan Africa (blue in the map), there are relatively high levels of no need (influenced by high fertility preferences amongst married adolescents), moderate levels of unmet need, and a low prevalence of contraceptive use.

Among unmarried women, there is a much greater concentration at high values of no need, with generally low contraceptive use and low unmet need. Some countries, whilst still having moderate to high levels of no need, have moderate levels of contraceptive use (in parts of Northern America and Europe, Southern Africa, South America) or unmet need for family planning (in Western and Middle Africa). The estimates of contraceptive prevalence, unmet

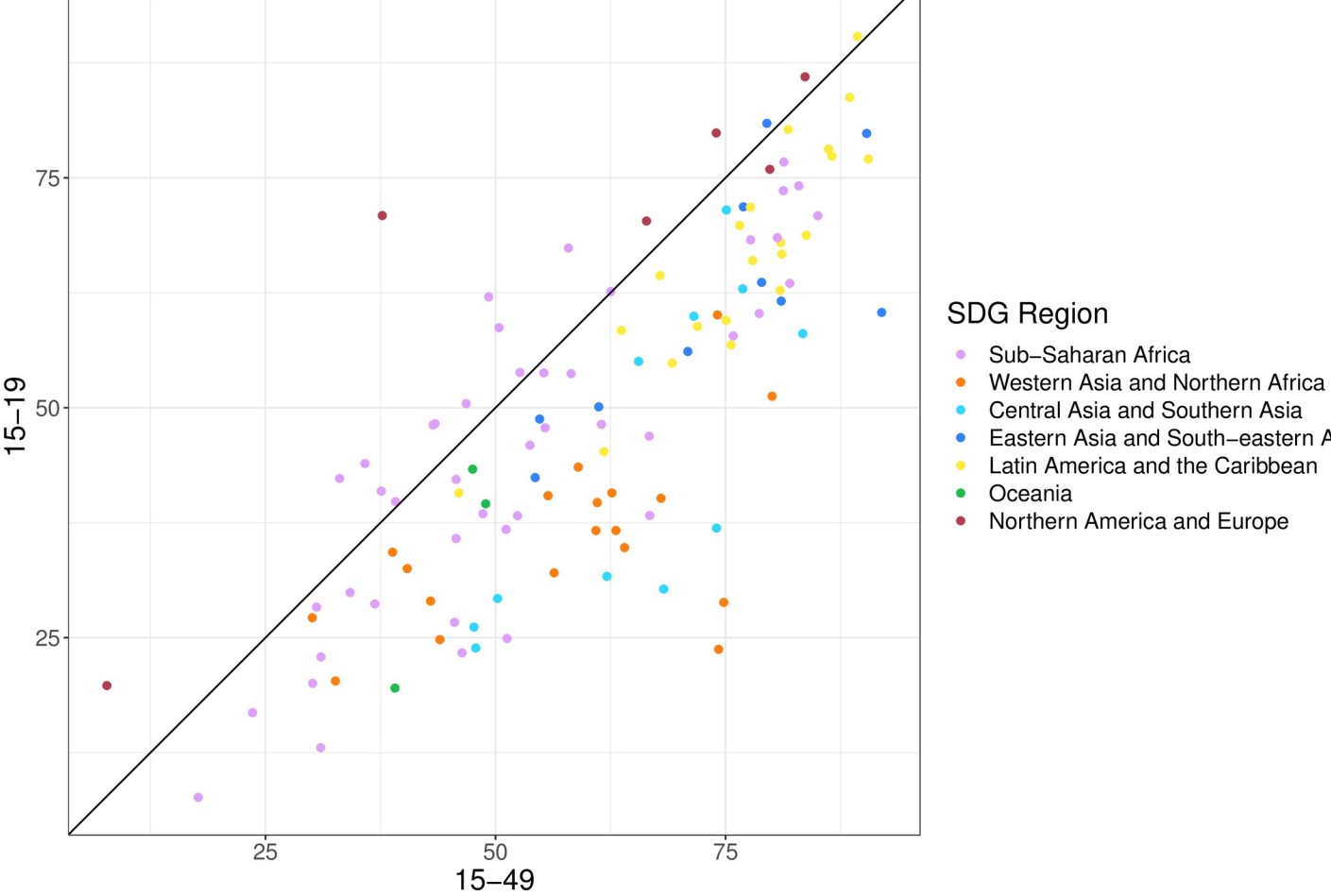

**Fig 6. Proportion of need for family planning satisfied by modern methods among women of reproductive age (15–49 years) and adolescent girls and women (15–19 years) in 2019, by country.** Note: The colours indicate regions.

need for, and no need for family planning amongst all adolescents (Fig 9) are influenced predominantly by unmarried women, since they contribute the greater share of all adolescents in nearly all countries of the world.

## Proportion of adolescents among all contraceptive users and women with unmet need for family planning

Among all women aged 15–49 globally, 15.3% were aged 15–19 (Table 5). These proportions are very different by marital status. While 38.8% of all unmarried women of reproductive age were 15–19, only 2.9% of married women of reproductive age were 15–19. In other words, the population of unmarried women of reproductive age is, demographically, much younger than the population of married women of reproductive age. Globally, adolescents constituted 7.9% of women experiencing unmet need for family planning and 3.2% among those using any method of contraception. More than one in ten women experiencing unmet need for family planning are adolescents in sub-Saharan Africa (13.3%) and Latin American and the Caribbean (14.0%). In every region and for every marital status category, the proportion of adolescents among all women experiencing unmet need for family planning is larger compared to the proportion of adolescent among all users of contraception.

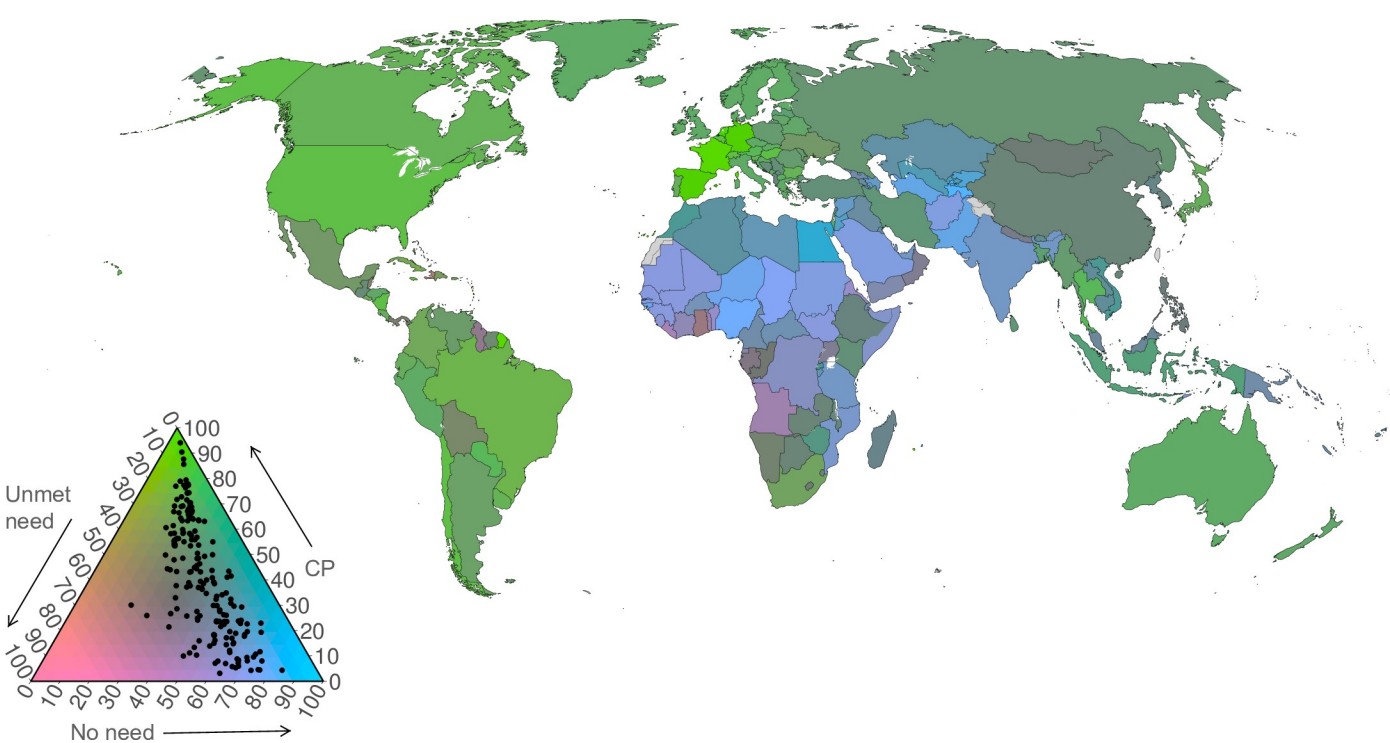

**Fig 7. Ternary colour scheme map for married adolescent women aged 15–19, showing posterior median estimates of contraceptive use (any method), unmet need for family planning, and no need for family planning in 2019.** The base map was obtained from Natural Earth (https://naturalearthdata.com). The boundaries and names shown and the designations used on this map do not imply official endorsement or acceptance by the United Nations. Note: The arrows point in the direction of increasing values.

## Numbers of contraceptive users and women with unmet need for family planning

The number of adolescent women aged 15–19 using contraception, experiencing a need for family planning, and having an unmet need for family planning, is given in Table 6 for year 2019, and Fig 10 and Supplementary information S2 and S3 Results for the period 1990 to 2030. Worldwide, over the period 1990 to 2019, the estimated number of adolescents in need of family planning increased, globally, from 33.3 million (95%UI = 28.4–57.9) to 44.8 million (95%UI = 39.0–63.3), and is projected to reach 51.1 million (95%UI = 43–72.7) in 2030. The apparent increase in need for family planning have been driven by the increase in users. In 2019, 29.8 million (95%UI = 24.6–41.7) adolescents were using any contraceptive method, double the number compared to 14.9 million (95%UI = 11.5–24.9) in 1990, while unmet need remained relatively stable among all adolescents, changing from 18.4 million (95%UI = 15.2–39.1) in 1990 to 15.0 million (95%UI = 12.1–29.2) in 2019.

Among 26.5 million (95%UI = 21.7–37.2) adolescents using modern methods of contraception, the number of unmarried adolescents using modern methods was greater than the number of users among married adolescents in 2019; globally, there were 9.4 million (95%UI = 8.6, 10.1) married adolescent users and 17.2 million (95%UI = 12.4, 27.7) unmarried adolescent users.

Among regions, the fastest increase in the number of adolescents with need for family planning was in sub-Saharan Africa and in 2019, one in four adolescents in need of family planning is living in sub-Saharan Africa.

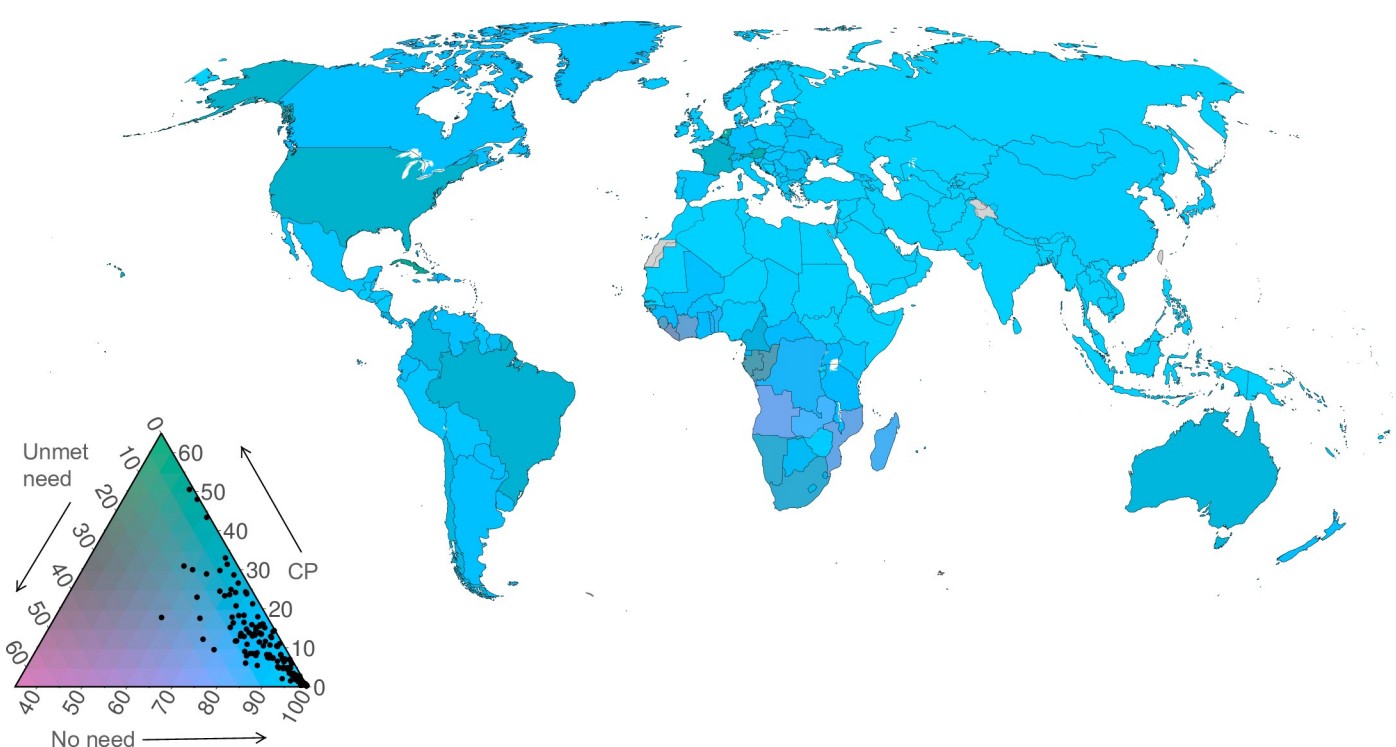

**Fig 8. Ternary colour scheme map for unmarried adolescent women aged 15–19 years in 2019, showing posterior median estimates of contraceptive use (any method), unmet need for family planning, and no need for family planning (percentages).** The base map was obtained from Natural Earth (https://naturalearthdata.com). The boundaries and names shown and the designations used on this map do not imply official endorsement or acceptance by the United Nations. Note: The arrows point in the direction of increasing values.

## Discussion

Globally, among adolescent women aged 15–19, it is estimated that 5.9% were using a method of contraception in 1990, and 10.2% in 2019. The proportion of unmarried women at these ages who were using some means of contraception increased from 3.8% in 1990 to 7.4% in 2019 and is projected to reach 8.6% in 2030. Amongst married adolescents, the proportion using a method of contraception increased from 15.2% in 1990 to 30.9% in 2019 and is projected to reach 37.2% by 2030. In 2019, the highest proportions of adolescents using contraception were found in Northern America and Europe (21.5% among unmarried, 71.4% among married), and Latin America and the Caribbean (18.9% among unmarried, 63.0% among married). In sub-Saharan Africa, the proportion of married women aged 15–19 who were using contraception in 2019 was relatively low, at 17.9%, while for unmarried adolescents it was just 9.4%.

Among adolescent women aged 15–19, 7.3% experienced an unmet need for family planning in 1990, and 4.8%, in 2019. Unmet need amongst married adolescents declined from 28.8% in 1990 to 20.7% in 2019 but is projected to decline more slowly between now and 2030, when it may lie around 19.5%. Among unmarried adolescents, the proportion having an unmet need for family planning was 2.4% in 1990 and 3.0% in 2019 and is projected to be around 3.3% in 2030. There was much greater variation in unmet need amongst unmarried compared to married adolescents across the regions. For example, unmarried adolescents in Western Asia and Northern Africa, and Central Asia and Southern Asia have very low levels of unmet need for family planning and use of contraception. This is because, according to available data, few unmarried adolescents in this region are sexually active and, therefore, it is estimated that the level of need for family planning is low in the population.

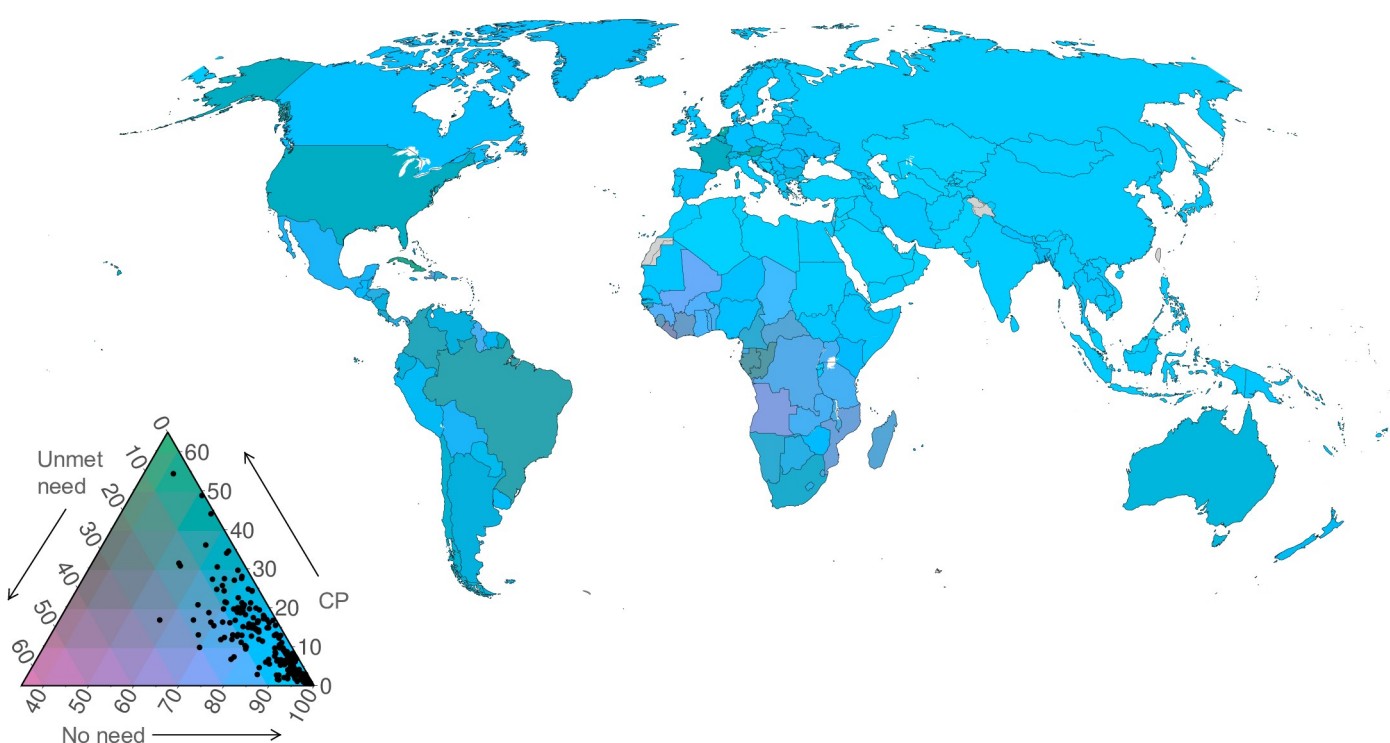

**Fig 9. Ternary colour scheme map for all adolescent women aged 15–19, showing posterior median estimates of contraceptive use (any method), unmet need for family planning, and no need for family planning in 2019.** The base map was obtained from Natural Earth (https://naturalearthdata.com). The boundaries and names shown and the designations used on this map do not imply official endorsement or acceptance by the United Nations. Note: The arrows point in the direction of increasing values.

Population growth and the postponement of marriage influence trends in the absolute number of adolescents using contraception or experiencing unmet need. Worldwide, the number of adolescent women aged 15–19 using any method of contraception increased from 14.9 million in 1990 to 29.8 million in 2019; it is projected to reach 35.9 million in 2030. The increase over time was driven by the increasing numbers of users among unmarried women. Among unmarried adolescent girls and young women at ages 15–19, the number using any method of contraception increased from 7.8 million in 1990 to 18.8 million in 2019; it is projected to reach 24.8 million in 2030. Among married women in the same age group, the number of contraceptive users increased from 7.1 million in 1990 to 11.0 million in 2019; it is projected to remain around 11.0 million through 2030.

The number of adolescent girls and young women at ages 15–19 who experience an unmet need for family planning declined from 18.4 million in 1990 to 15.0 million in 2019; it is projected to remain around 15 million in 2030. Trends in this indicator by marital status are moving in opposite directions. The number of unmarried adolescent women who have an unmet need for family planning grew from 4.9 million in 1990 to 7.6 million in 2019; it is projected to reach 9.5 million in 2030. The same indicator for married adolescents, however, declined from 13.4 million in 1990 to 7.4 million in 2019; it is projected to decline further to 5.8 million in 2030.

In some regions the increase in contraceptive prevalence among married adolescent women was offset by a declining proportion of adolescents who were married. In countries where contraceptive use and the need for family planning are projected to remain relatively low among unmarried adolescent women, the total number of contraceptive users at ages 15–19 could decline despite the increased use of contraception among those who are married.

**Table 5. Proportion of women of reproductive age (15–49 years) who are aged 15–19 among those using some method of contraception, experiencing an unmet need for family planning, and having a need for family planning (in per cent), by region, 2019.**

| Marital status | Region | Proportion of women (in per cent) aged 15–19 among those aged 15–49 . . . | | | |
|---|---|---|---|---|---|
| | | . . . using some method of contraception | . . . experiencing an unmet need for family planning | . . . having a need for family planning | . . . in the population |
| All women | World | 3.2 | 7.9 | 4.0 | 15.3 |
| | Sub-Saharan Africa | 8.7 | 13.3 | 10.4 | 22.2 |
| | Western Asia and Northern Africa | 1.6 | 5.0 | 2.3 | 16.5 |
| | Central and Southern Asia | 1.6 | 4.3 | 2.2 | 16.6 |
| | Eastern and South-eastern Asia | 1.2 | 4.9 | 1.6 | 12.1 |
| | Latin America and the Caribbean | 6.7 | 14.0 | 7.6 | 15.4 |
| | Oceania | 5.7 | 6.4 | 5.8 | 14.7 |
| | Northern America and Europe | 4.8 | 6.7 | 5.0 | 12.0 |
| Married | World | 1.4 | 4.7 | 2.0 | 2.9 |
| | Sub-Saharan Africa | 4.0 | 7.8 | 5.5 | 7.4 |
| | Western Asia and Northern Africa | 1.2 | 3.4 | 1.7 | 2.8 |
| | Central and Southern Asia | 1.5 | 4.4 | 2.2 | 3.4 |
| | Eastern and South-eastern Asia | 0.6 | 2.4 | 0.8 | 1.0 |
| | Latin America and the Caribbean | 3.3 | 7.6 | 3.8 | 3.9 |
| | Oceania | 1.1 | 2.4 | 1.3 | 1.7 |
| | Northern America and Europe | 0.9 | 1.1 | 0.9 | 0.9 |
| Unmarried | World | 13.1 | 23.4 | 15.0 | 38.8 |
| | Sub-Saharan Africa | 20.8 | 36.3 | 25.3 | 45.6 |
| | Western Asia and Northern Africa | – | – | – | 37.3 |
| | Central and Southern Asia | – | – | – | 51.6 |
| | Eastern and South-eastern Asia | 6.8 | 13.3 | 8.0 | 38.0 |
| | Latin America and the Caribbean | 15.4 | 27.6 | 17.1 | 30.0 |
| | Oceania | 15.7 | 21.6 | 16.5 | 32.4 |
| | Northern America and Europe | 12.3 | 17.8 | 12.9 | 24.7 |

To our knowledge, this is the first time that estimates and projections of key family planning indicators for adolescent women aged 15 to 19 have been produced covering such a large number of countries and regions and the period from 1990 to 2030.

The experiences of girls and young women aged 15 to 19 differ considerably by marital status. For this reason, the estimates and projections were prepared and presented separately by marital status, in order to provide a more nuanced picture of contraceptive use and unmet need. Additionally, they take into account changing patterns of marriage and union formation over time, in particular the decline in the proportion married. The model also accounts for

**Table 6. Estimates and projections of the number of users (millions) of contraception (any method and modern methods), number experiencing an unmet need for family planning, and number in need of family planning among women aged 15–19 years, by marital status, 2019, for regions.**

| Marital status | Region | Contraceptive users, any method (million) | Contraceptive users, modern methods (million) | Need for family planning (million) | Unmet need for family planning (million) |
|---|---|---|---|---|---|
| All women | World | 29.80 (24.60, 41.70) | 26.50 (21.70, 37.20) | 44.80 (39.00, 63.30) | 15.00 (12.10, 29.20) |
| | Sub-Saharan Africa | 6.37 (5.74, 7.14) | 5.57 (5.02, 6.23) | 12.20 (11.30, 13.30) | 5.85 (5.26, 6.56) |
| | Western Asia and Northern Africa | 0.72 (0.54, 2.72) | 0.51 (0.38, 1.80) | 1.33 (1.04, 3.52) | 0.61 (0.41, 1.63) |
| | Central and Southern Asia | 3.51 (2.96, 6.47) | 2.71 (2.24, 4.56) | 6.23 (5.53, 9.25) | 2.71 (2.18, 3.39) |
| | Eastern and South-eastern Asia | 4.19 (2.19, 15.00) | 3.69 (1.93, 13.70) | 5.92 (3.35, 23.90) | 1.73 (0.72, 15.30) |
| | Latin America and the Caribbean | 6.79 (5.00, 9.62) | 6.37 (4.62, 9.14) | 8.72 (6.89, 11.50) | 1.93 (1.25, 3.41) |
| | Oceania | 0.27 (0.09, 0.53) | 0.26 (0.08, 0.52) | 0.34 (0.14, 0.74) | 0.07 (0.02, 0.44) |
| | Northern America and Europe | 6.92 (4.54, 9.71) | 6.55 (4.23, 9.26) | 8.08 (5.60, 13.00) | 1.16 (0.35, 6.03) |
| Married | World | 11.00 (10.20, 11.90) | 9.36 (8.63, 10.10) | 18.40 (17.50, 19.40) | 7.39 (6.68, 8.20) |
| | Sub-Saharan Africa | 2.10 (1.89, 2.33) | 1.81 (1.61, 2.02) | 4.88 (4.57, 5.24) | 2.79 (2.51, 3.10) |
| | Western Asia and Northern Africa | 0.52 (0.43, 0.64) | 0.38 (0.30, 0.47) | 0.92 (0.80, 1.06) | 0.40 (0.31, 0.50) |
| | Central and Southern Asia | 3.21 (2.68, 3.82) | 2.53 (2.06, 3.06) | 5.88 (5.21, 6.66) | 2.67 (2.13, 3.32) |
| | Eastern and South-eastern Asia | 1.88 (1.53, 2.26) | 1.69 (1.35, 2.06) | 2.53 (2.21, 2.84) | 0.65 (0.45, 0.92) |
| | Latin America and the Caribbean | 2.43 (2.13, 2.70) | 2.21 (1.92, 2.47) | 3.14 (2.95, 3.30) | 0.71 (0.54, 0.92) |
| | Oceania | 0.03 (0.02, 0.05) | 0.03 (0.02, 0.04) | 0.06 (0.04, 0.07) | 0.02 (0.01, 0.04) |
| | Northern America and Europe | 0.83 (0.74, 0.91) | 0.71 (0.60, 0.81) | 0.96 (0.89, 1.03) | 0.13 (0.08, 0.20) |
| Unmarried | World | 18.80 (13.60, 30.70) | 17.20 (12.40, 27.70) | 26.40 (20.60, 44.90) | 7.59 (4.90, 21.70) |
| | Sub-Saharan Africa | 4.27 (3.71, 4.99) | 3.76 (3.26, 4.37) | 7.34 (6.54, 8.30) | 3.07 (2.62, 3.64) |
| | Western Asia and Northern Africa | — | — | — | — |
| | Central and Southern Asia | — | — | — | — |
| | Eastern and South-eastern Asia | 2.30 (0.38, 13.10) | 2.00 (0.30, 12.00) | 3.39 (0.83, 21.30) | 1.08 (0.13, 14.70) |
| | Latin America and the Caribbean | 4.36 (2.63, 7.15) | 4.16 (2.48, 6.91) | 5.58 (3.79, 8.33) | 1.22 (0.58, 2.66) |
| | Oceania | 0.24 (0.06, 0.50) | 0.23 (0.05, 0.49) | 0.29 (0.09, 0.68) | 0.05 (0.00, 0.42) |
| | Northern America and Europe | 6.09 (3.73, 8.88) | 5.83 (3.54, 8.54) | 7.11 (4.64, 12.00) | 1.03 (0.22, 5.90) |

Note: Results are 'median (lower, upper)', where 'median' is the posterior median and 'lower' and 'upper' are the lower and upper limits of the posterior 95% uncertainty intervals.

misclassification errors, and other biases such as non-standard age groups, through bias and misclassification parameters.

In cases where there are limited or no data, prediction intervals are especially useful. For the age group 15–19 years, there are limited data and therefore the uncertainty around crucial determinants of family planning use, such as sexual activity amongst unmarried adolescents, is large. Presentation of uncertainty intervals can help to communicate the plausible range of outcomes.

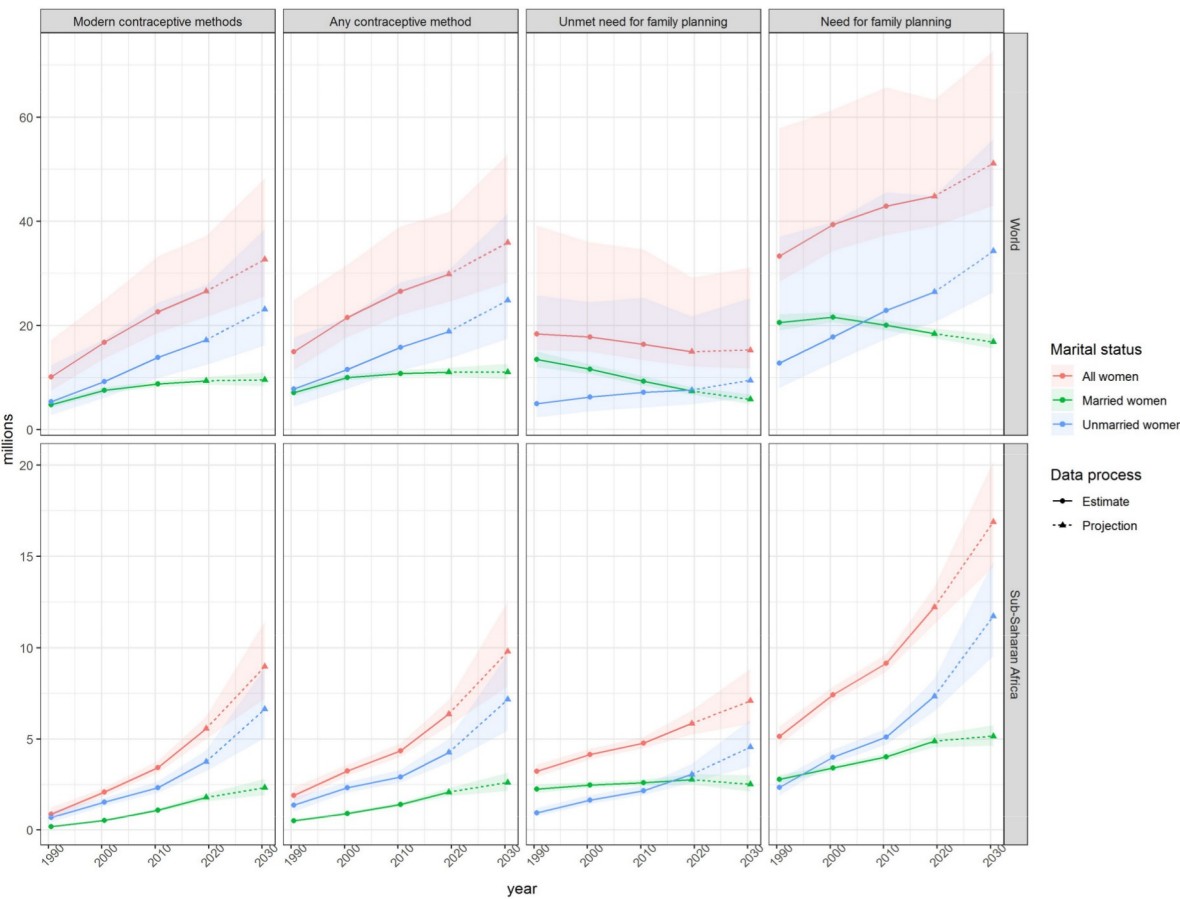

**Fig 10. Estimates and projections the number of users (millions) of contraception (any method and modern methods), number experiencing unmet need for family planning, and number in need of family among women aged 15–19, by marital status, 2019, for the world and sub-Saharan Africa.** Given are the posterior medians (solid lines) posterior 95% uncertainty intervals (ribbons).

There are several limitations of the estimates of family planning indicators among adolescents presented here. Uncertainty in estimates and projections around all indicators was high for Oceania, Eastern and South-Eastern Asia, Northern America and Europe, and Latin America and the Caribbean, particularly for unmarried women. Large uncertainty intervals are caused by the low availability of data for countries in these regions, presence of sampling biases by age, and by regional heterogeneity.

The density of the data cloud in the time period of estimation also affects the magnitude of the uncertainty. For example, the uncertainty interval for worldwide unmet need for family planning among unmarried women in 1990 was (95%UI = 1.2–12.6), a width of 11.4 percentage points. In 2019, it was (95%UI = 1.9–8.5), a width of 6.6 percentage points. There were more observations available in the decade preceding 2019 compared to previous decades.

Some limitations of household survey data may have affected the model-based estimates and projections presented here. For instance, some surveys are not representative of the population of women aged 15–19, as they miss those adolescents schooled elsewhere and living away from home, only interview women aged 18 and older, or have reduced coverage for

other reasons. While some of these biases are addressed by the inclusion of bias parameters in the model, the result is a concomitant increase in uncertainty.

Furthermore, adolescents, particularly those who are unmarried, are likely to under-report sexual activity and contraceptive use. It is assumed that this is because they may feel uncomfortable answering questions regarding their sexual behaviour, which in some settings is stigmatised. Studies have shown that the mode of interviewing and questions asked affected the reporting of sexual activity among adolescents, though not always in accordance with expectations [32,33].

In order to calculate unmet need for family planning, it was assumed that all married adolescents were sexually active (except those who were pregnant or amenorrhoeic). Amongst unmarried women, we followed the approach used with the Demographic and Health Surveys of defining an unmarried adolescent as being currently sexually active if she had had sex within 28 days before the survey. Our results would have differed if we had chosen a different criterion. For example, if we had chosen a criterion of sex within three months or one year before the survey, or of ever having had sex, this would have resulted in higher estimates of unmet need for family planning amongst unmarried adolescents.

We displayed country results on choropleth world maps with ternary plots and ternary-balance color schemes. These provided a general comparison of family planning compositions across marital groups in only two maps. While efficient in terms of information per map, these displays do require greater effort to interpret on behalf of the reader. Moreover, as with standard choropleth maps, information about uncertainty is not included; the reader would have to consult the tables also provided to retrieve this.

A final potential limitation is that the statistical model was applied to data for the age group 15–19 years in isolation. Similar estimates are produced for all women of reproductive age [11] and further work in the pooled estimation of all seven 5-year age group of reproductive age (15–49 years) is needed using modelling techniques for compositional data [34].

In monitoring the Sustainable Development Goals of the 2030 Agenda, data disaggregated along meaningful dimensions can be used to identify sub-groups that are being left behind. As the family planning community is committed to ensuring that girls and women of all ages and in all countries have access to sexual and reproductive health-care services, it is crucial to understand the situation of adolescents who typically face additional barriers to receiving care. We show that the proportion of need for family planning satisfied with modern methods (SDG indicator 3.7.1.) is lower among adolescents compared to other reproductive ages. Previous studies have highlighted what is needed, such as informing youth about available sources of contraceptives and improving their access to them, especially by reducing social barriers [35]. Young women in sub-Saharan Africa used more short-term contraceptive methods through private providers compared with older women. Therefore, interventions to increase adolescents' access to a range of methods and competent counselling should target providers frequently used by young people, including providers in the private sector [36]. Additionally, in some settings, improving access to and the quality of family-planning services may not markedly increase contraceptive use among adolescents and young women without broader shifts in norms regarding early childbearing [37]. The estimates and projections of family planning indicators presented here can provide insights into where investments in family planning programmes are needed, in order to adequately provide for the needs of girls and young women.

## Supporting information

**S1 Checklist. GATHER checklist.**
(DOCX)

**S1 Appendix. Technical appendix providing supplementary information on input data and methodology, plus supplementary results.**
(PDF)

**S2 Appendix. Supplementary figures; country specific estimates and projections of family planning indicators.**
(PDF)

**S1 Data. Table C in S1 Appendix (country and region classifications) provided in csv format.**
(CSV)

**S2 Data. Proportion of women aged 15–19 who are married, by subregion, development group and income group.**
(XLSX)

**S3 Data. The input data file, including biases and misclassifications.**
(CSV)

**S1 Results. Contraceptive prevalence (any method, modern methods), the unmet need for family planning, the proportion of need for family planning satisfied by modern contraceptive methods (SDG indicator 3.7.1), and total need for family planning, by SDG country groupings at level 1.**
(XLSX)

**S2 Results. Contraceptive prevalence (any method, modern methods), the unmet need for family planning, the proportion of need for family planning satisfied by modern contraceptive methods (SDG indicator 3.7.1), and total need for family planning, by SDG country groupings at level 2.**
(XLSX)

**S3 Results. Contraceptive prevalence (any method, modern methods), the unmet need for family planning, the proportion of need for family planning satisfied by modern contraceptive methods (SDG indicator 3.7.1), and total need for family planning, by country income levels.**
(XLSX)

**S4 Results. Contraceptive prevalence (any method, modern methods), the unmet need for family planning, the proportion of need for family planning satisfied by modern contraceptive methods (SDG indicator 3.7.1), and total need for family planning, by development groups.**
(XLSX)

**S1 File.**
(PDF)

## Acknowledgments

The views expressed in this paper are those of the authors and do not necessarily reflect the views of the UN.

## Author Contributions

**Conceptualization:** Vladimíra Kantorová, Mark C. Wheldon, Aisha N. Z. Dasgupta, Philipp Ueffing, Helena Cruz Castanheira.

**Data curation:** Vladimíra Kantorová, Aisha N. Z. Dasgupta, Philipp Ueffing, Helena Cruz Castanheira.

**Formal analysis:** Vladimíra Kantorová, Mark C. Wheldon, Philipp Ueffing.

**Funding acquisition:** Vladimíra Kantorová.

**Investigation:** Aisha N. Z. Dasgupta.

**Methodology:** Vladimíra Kantorová, Mark C. Wheldon, Aisha N. Z. Dasgupta, Philipp Ueffing, Helena Cruz Castanheira.

**Project administration:** Vladimíra Kantorová.

**Resources:** Vladimíra Kantorová.

**Software:** Mark C. Wheldon.

**Supervision:** Vladimíra Kantorová.

**Validation:** Mark C. Wheldon.

**Visualization:** Mark C. Wheldon, Philipp Ueffing.

**Writing – original draft:** Vladimíra Kantorová, Mark C. Wheldon, Aisha N. Z. Dasgupta, Philipp Ueffing, Helena Cruz Castanheira.

**Writing – review & editing:** Vladimíra Kantorová, Mark C. Wheldon, Aisha N. Z. Dasgupta, Philipp Ueffing, Helena Cruz Castanheira.

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
