## [Decision Letter · Decision Letter 0]

29 Oct 2020

PONE-D-20-16904

Contraceptive Use and Needs among Adolescent Women Aged 15-19:  

Regional and Global Estimates and Projections from 1990 to 2030 from a Bayesian Hierarchical Modelling Study

PLOS ONE

Dear Dr. Kantorová,

Thank you for submitting your manuscript to PLOS ONE. After careful consideration, we feel that it has merit but does not fully meet PLOS ONE’s publication criteria as it currently stands. Therefore, we invite you to submit a revised version of the manuscript that thoroughly addresses all points raised by both reviewers.

We look forward to receiving your revised manuscript.

Kind regards,

Philip Anglewicz, PhD

Academic Editor

PLOS ONE

Journal Requirements:

2. We note that Figures 7, 8, 9 in your submission contain map images which may be copyrighted. All PLOS content is published under the Creative Commons Attribution License (CC BY 4.0), which means that the manuscript, images, and Supporting Information files will be freely available online, and any third party is permitted to access, download, copy, distribute, and use these materials in any way, even commercially, with proper attribution. For these reasons, we cannot publish previously copyrighted maps or satellite images created using proprietary data, such as Google software (Google Maps, Street View, and Earth). For more information, see our copyright guidelines: http://journals.plos.org/plosone/s/licenses-and-copyright.

2.1.    You may seek permission from the original copyright holder of Figures 7, 8, 9 to publish the content specifically under the CC BY 4.0 license. 

2.2.    If you are unable to obtain permission from the original copyright holder to publish these figures under the CC BY 4.0 license or if the copyright holder’s requirements are incompatible with the CC BY 4.0 license, please either i) remove the figure or ii) supply a replacement figure that complies with the CC BY 4.0 license. Please check copyright information on all replacement figures and update the figure caption with source information. If applicable, please specify in the figure caption text when a figure is similar but not identical to the original image and is therefore for illustrative purposes only.

Reviewers' comments:

Reviewer's Responses to Questions

**Comments to the Author**

1. Is the manuscript technically sound, and do the data support the conclusions?

Reviewer #1: Yes

Reviewer #2: Yes

2. Has the statistical analysis been performed appropriately and rigorously? 

Reviewer #1: I Don't Know

Reviewer #2: Yes

3. Have the authors made all data underlying the findings in their manuscript fully available?

Reviewer #1: Yes

Reviewer #2: Yes

4. Is the manuscript presented in an intelligible fashion and written in standard English?

Reviewer #1: Yes

Reviewer #2: Yes

5. Review Comments to the Author

Reviewer #1: I think this study could constitute a valuable and needed contribution to the literature. The authors’ approach allows for comparison of trends over time; is global in scope; represents a massive compilation of data; and they estimated uncertainty intervals which can help readers interpret and understand differences between regions/countries and over time.

I also think this study could be strengthened in several areas.

Please see my comments in the attached file.

Reviewer #2: This is an excellent paper that assembles lots of data and presents a careful analysis. There will be lots of interest in the results and the methods to arrive at those results. I have only a few minor issues to address.

1. Lines 223-230. I could not understand the differences between the first sentence of the paragraph ‘For each country, the model generates estimates of …’ and the third sentence ‘In the modeling process…’ It seems to me that both sentences mean the same.

2. Lines 280-281. The paper states that Group 0 is composed of countries with ‘very low’ rates of sexually activity among adolescent women, but ‘very low’ is never defined. Can you please provide the cut-off value used?

3. Line 369. ‘Compositional indicators were formed from the posterior medians…’ You cannot expect the medians of the four groups to sum to one (unlike the averages) so how did you derive the compositional indicators? Did you just normalize the medians?

4. Line 423. I did not understand this sentence: ‘In these two regions, the estimated level of sexual activity among unmarried women in general, and among unmarried adolescents in particular, is low, accounting for the low levels of sexual activity, need for family planning and use of contraception.’ Perhaps you just need to remove ‘sexual activity’ from the second part of the sentence to avoid the tautology.

5. At first glance, I thought the ternary diagrams and colored maps where going to be very hard to interpret. But they actually worked reasonably well. This is an interesting approach to presenting three-dimensional information.

6. I found it a bit cumbersome to think of the third component of the compositional indicator as ‘no need’ since it is not apparent to me how it should vary by region. But if I thought of it as ‘low levels of sexual activity’ then it was easy to follow. I assume that most of the ‘no need’ category is composed of women who are not sexually active, but there must be some component of those who want another child now. Could you show the proportion of the ‘no need’ group that comes from each component? It would certainly help to understand the indicator better.

6. PLOS authors have the option to publish the peer review history of their article (what does this mean?). If published, this will include your full peer review and any attached files.

Reviewer #1: No

Reviewer #2: No

---

## [Author Response · Author response to Decision Letter 0]

15 Dec 2020

Reviewer #1: 

I think this study could constitute a valuable and needed contribution to the literature. The authors’ approach allows for comparison of trends over time; is global in scope; represents a massive compilation of data; and they estimated uncertainty intervals which can help readers interpret and understand differences between regions/countries and over time.

Thank you for these comments.

Methods

The section on statistical methods begins 

A Bayesian hierarchical model was used to estimate and project contraceptive prevalence and the unmet need for family planning among women aged 15-19 years. The estimates for women who are married or in a union were derived separately from those for women who are not married or in a union, using methods developed for estimating and projecting family planning indicators among all women of reproductive age (15-49 years) [19-21].

The paragraph cites Alkema et al (2013), the Lancet paper which describes how contraceptive needs and use were estimated for married women, to a review of survey data, and Wheldon et al (2018), a UNPD technical paper which describes how that model was revised for unmarried women. I assume from this, and also because the manuscript doesn't include technical details on the modeling approach, that these citations were to convey that the authors applied the same methodology but to a subset of those data. However, this should be clarified. I.e., please specify whether the methods were exactly the same except for the data restrictions (or if not, please specify what the differences were).

We used the same statistical model but applied it to the input data on women aged 15–19 years. We have updated the last few lines of the first paragraph of Section “Statistical Methods”, Subsection “Modelling trends in contraceptive use and unmet need for family planning”. It now reads:

… We used the same model for estimating and projecting family planning indicators developed in earlier research for all women of reproductive age (15-49 years) [19-21], except we applied it to data only on women aged 15-19 years. We refer the reader to the supplementary material of [21] for the full model specification; we give a conceptual overview below.

I think it’s probably fine to apply this model. However, the authors have not presented out-of-sample validation exercises. These are needed to assess how well the model performs for years lacking reliable data and for countries for which data are unavailable, and to assess the forward-projections.

We performed out-of-sample model validation exercises and reported them in “S1 Appendix” (methodology and results). These show that the model fits the data well and has good out-of-sample predictive validity. The following text was added near the end of Subsection “Parameter Estimation and Software” in Section “Statistical Methods”:

Out-of-sample validation exercises were performed to assesses model fit; these are reported in S1 Appendix. The exercises indicated that the model fitted the data well and had good out-of-sample predictive validity.

The authors define sexually active as having had intercourse within the past four weeks. The authors state that they used four-week intervals because those are “comparable to those generally published in the survey reports.” However, this definition strikes me as closer to having recently had sex than sexually active. In contrast, the Adding It Up study defines sexually active as having had sex within the past three months. The other analysis which the authors cited, Behrman et al (2018), uses as the denominator women who have ever had sex. Both of these studies used DHS data. Among the reasons a 28-day window surprised me is that this would result in smaller sample sizes in contrast to a 3-month window, exacerbating the data limitations. Another concern I have is whether this shorter interval is more likely to capture individuals who have sex more frequently. Their contraceptive behaviors could differ, and this could bias the estimates of demand satisfied.

Yes, we agree that the window of 28 days might be too narrow to define who is ‘sexually active’. Both definitions (28-days and 3-months) have advantages and shortcomings, and we decided to follow DHS definition for a greater transparency of data used in the estimates. Additionally, the newest Adding-It-Up from July 2020, available at https://www.guttmacher.org/report/adding-it-up-investing-in-sexual-reproductive-health-2019, also uses the shorter window.

We agree that the sample size for the number of unmarried women in need for family planning is very small in many surveys. Therefore, we do not present results for the demand for family planning satisfied with modern methods (SDG 3.7.1) among unmarried women for low sexual activity group of countries (and for SDG regions of Northern Africa and Western Asia, and Central and Southern Asia).

The authors write that for married women, countries were clustered within subregions, and subregions, in turn, were clustered within regions. In contrast, for unmarried women, subregions were not clustered within regions. Rather, subregions were clustered into a very-low-sex cluster or an everywhere-else cluster. If I follow, then, this would mean that, for married women, the subregional parameters for, e.g., Western Africa, would be centered around an Africa mean, and for Western Europe, centered around a Europe mean, but, for unmarried women, the subregional parameters for Western Africa and Western Europe would be centered around the same mean because they're both grouped in the everywhere-else region.

Yes, this is correct. To make the hierarchies clearer we have added extra information, including diagrams, in the new “S1 Appendix”.

The authors explain that they clustered countries in this way because, "For unmarried adolescent women, need for family planning is closely related to level of sexual activity and sexual activity among unmarried adolescent women varies considerably between countries." However, sexual activity -- i.e. whether a women is in the "not in need" group or not -- is an unknown that they're modeling, and it's in their data.

Women in the “not in need” category among unmarried women may be either sexually inactive, or sexually active who want to have a child, are infecund or are pregnant or postpartum amenorrhoeic (when the pregnancy was intended). Vast majority of “no need” category among unmarried women in age group 15-19 are sexually inactive (defined as no sexual intercourse in past 28 days) thus the no need for family planning is related to the level of sexual activity. The figure below shows the proportions of sexually inactive among unmarried women who have no need for family planning, by woman’s age, from 186 DHS and MICS surveys (results are shown as a box plot of distribution and including outliers).

“Not in need” category is modelled in the same way as other categories of contraceptive use and unmet need for family planning using all survey data available. 

To clarify the diversity of the group of “not in need” we add a text in Definitions:

Women who are not in need of family planning are women who are not contraceptive users and are not classified as having an unmet need for family planning. For unmarried women, this group includes women who are either sexually inactive (as defined by no sexual intercourse in past 28 days), or sexually active who want to have a child, are infecund or are pregnant or postpartum amenorrhoeic (when the pregnancy was intended). Most unmarried women aged 15-19 years who have no need for family planning are sexually inactive.

Reviewing the technical paper that they cited, Figure 7 on page 11 of Wheldon et al appears to show that "very low" sexual activity roughly corresponds to the Middle East and North Africa, and to LMIC Asia minus China. In several of those countries, my understanding is that data for unmarried women's sexual activity are often unavailable, particularly in earlier periods.

That the DHS doesn't ask these questions in several countries could reflect greater stigma about sexual activity. This could predict lower levels of sexual activity, and also differences in access to and use of contraception among those who are sexually active. Thus, given the nonrandom availability of data for unmarried women, I can see how exchanging information across countries using a geographic regional schema could result in biased estimates.

We used available data to classify countries according to sexual activity among unmarried women, including data from DHSs, MICSs and other surveys in 81 countries that included questions on sexual activity among unmarried women. In countries where these questions were not included, information on the acceptance of sex between unmarried adults and the level of religiosity was used as a proxy. This information came from Pew Research Center’s Global Attitudes Survey (2014) and Global Religious Landscape survey (2012), and the World Values Survey (round 6). We believe that using this additional information improves the assignment of countries into two clusters, in particular for countries in Western Asia and Northern Africa where surveys like DHS or MICS do not ask the questions on sexual activity and contraceptive use among unmarried women. 

We included this information in the section on the hierarchical structures in the new “S1 Appendix”. In the text of the main paper, we modified the end of Subsection “Hierarchical structures of models for married and unmarried women” in Section “Statistical Methods” to read:

Group 0 consisted of countries where sexual activity among unmarried women was estimated to be very low, defined as the proportion of unmarried women of reproductive age who report sexual activity in past 28 days was not more than 2 percent. Group 1 consisted of all other countries. To estimate the extent of sexual activity among unmarried women was estimated from the question on recent sexual activity in DHS, MICS and other available surveys. In countries without this data, we used information about the acceptance of sex between unmarried adults from other surveys and the level of religiosity. For further details on the data and methods used, see S1 Appendix. 

As noted in our Discussion (orig. submission, lines 656–660), we accept that there may be some bias due to a reluctance to respond to questions related to recent sexual activity and current contraceptive use, or a tendency to give inaccurate answers. 

However, as written, the authors' description of the clustering schema for unmarried women does not address these points, and it could come across as grouping countries based on the values of one of the dependent variables that they are estimating. For these reasons, I think the manuscript would be improved were these issues explained in the manuscript, in contrast to how the clustering schema is described in the text as presently written.

As noted above, we have added extra information on the hierarchical structures in the new “S1 Appendix” regarding assignment to one of the two sexual activity groups, and how it is based on data additional to that used to estimate family planning. The clusters do not directly constraint the estimated proportions of women of using contraception, having unmet need for family planning or no need. It means that in countries with little-to-no data the proportions of “not in need” is not constrained by the threshold of sexual activity used to define the sexual activity classification. Rather, the classification merely weights the data that do exist such that data from countries in the same sexual activity group are more informative.

We acknowledge that the classification scheme is crude in that it only had two levels, but the validation results (now included in the S1 Supplementary information) indicated good fit of the model. 

It's possible than an underlying issue is that the UNPD regional groupings weren't designed with modeling in mind. I suspect that using GBD regions and super-regions would make more sense for the authors' models. This could make it possible for the model hierarchy to be the same for married women and for unmarried women.

We appreciate the suggestion, but we do not think it would make a substantial difference switching to a different geographic classification such as GBD regions. The model validation exercises we have added (see below) showed the model fit is good for the chosen hierarchies. As discussed in the manuscript and the new appendix (“S1 Appendix”), we devised the sexual activity grouping to explicitly account for different levels of sexual activity among unmarried women, and used additional data to determine country classifications. As far as we are aware, the GBD regions were not designed to reflect different family planning or sexual activity characteristics and so would probably not provide a suitable clustering for family planning estimates. 

Related to this, it would be very valuable to perform sensitivity analyses around aspects of the model which are subjective. At a minimum, it would be informative to compare model validation exercises using alternate clustering schemas (e.g., subregions grouped within regions; subregions grouped by sexual activity; GBD regions and super-regions), and to what extent are the estimates of contraceptive needs and use are affected by alternate sexual activity windows.

We selected the geographic and sexual activity clustering schemes because they performed well for the 15–49 age group (Kantorová., et al. 2020, PLOS Medicine, 17(2), e1003026) and were based on substantive reasoning about factors (sexual activity among unmarried women and related population attitudes and opinions) likely to have an impact on contraceptive use. Kantorová, et al. (2020, PLOS Medicine, 17(2), e1003026) performed a sensitivity analysis for unmarried women of reproductive age (15–49 years) by re-running the model with Indonesia moved from the ‘low’ to the ‘other’ sexual activity group but found no meaningful change in the results. 

Our validation exercises (now included in S1 Appendix) show that current model and hierarchical structures also perform well for the 15–19 age group. There are many different ways one could design the hierarchical structure but testing many different options would be very time consuming, especially if the validation exercises are repeated in each case. Given our validation results showed that out-of-sample coverage of the uncertainty intervals was very close to the nominal values we feel there would be little to gain in terms of predictive accuracy from a lengthy exploration of other possible clusters, especially if there aren’t strong substantive reasons to guide the search. 

The major step in further improvement of the estimates will be possible only when more survey data become available. This would also reduce the uncertainty intervals around estimates in countries with no or limited data.

We added additional text in S1 Appendix, subsection 2.1:

While it might be possible to explore different hierarchical structures, out-of-sample validation exercises (see Sections 2.2 and 3.3) indicated the structure defined here performed well in terms of predictive accuracy. Therefore, we would expect any improvements under other structures to be limited.

Results

The "need for family planning" indicates the proportion of adolescent women who are sexually active and do not want to have a(nother) child. Among these, it is further informative to know the demand satisfied -- i.e., the percent of those "in need" who are using a (modern) method of contraception. 

However, the results section begins with contraceptive prevalence. Since levels of sexual activity differ across countries and change over time, it's not clear to me how to interpret this. 

The results section then discusses "unmet need". However, a country could have higher levels of unmet need because more people are having sex, while at the same time a smaller proportion of those having sex (who do not want to have a(nother) child) could have an unmet need for contraception. For these reasons, I think the authors need to do more to explain how this indicator should be interpreted. I can imagine that its utility could be that while it obscures sexual activity and demand satisfied, it predicts population-level adolescents unintended pregnancy among. 

The results section then discusses "need for family planning", and, after that, demand satisfied ("need for family planning satisfied by modern methods").

I think it would make more sense if the results section first discussed "need for family planning," then "need for family planning satisfied by modern methods", and then "unmet need for family planning." This is because order to understand comparisons, I think a reader needs to first understand differences in sexual activity levels. The discussion demand satisfied should also, instead of just describing differences in demand satisfied, contextualize these differences. One way to do this is to describe need; then demand satisfied; and then in a third subsection to explain how these two combine to produce unmet need.

Thank you for the suggestion on the structure of the results section. We discussed the best structure to present the results for four main indicators - contraceptive use, unmet need for family planning, total need for family planning and the demand for family planning satisfied with modern methods (SDG 3.7.1.). In the original submission, we opted to start with contraceptive use, since this is the mostly commonly used indicator of family planning. However, after reading your comments, we revised the structure of the result section and start with “need for family planning” and discuss how among unmarried women these are related to different levels of sexual activity and among married women to pregnancy intentions and expectations within marriage.

The authors also include ternary maps which blend three hues to jointly illustrate % unmet need + % using contraception + % no need. These maps show results by country. However, the authors do not discuss country results. They only discuss subregional results. The maps also obscure differences across countries in the uncertainty in the country estimates. The maps are also difficult to read, and it could be more useful if presenting maps to illustrate heat maps which report just one indicator. E.g., a map of % in need + a map of % demand satisfied. However, given that the authors don't actually discuss country results in their manuscript, country maps might not actually be appropriate to include alongside the main text.

We appreciate that the ternary maps contain a lot of information but we do feel that they provide a useful insight into the different multivariate configurations of the family planning components and show the main global patterns and differences. 

We also took into account that Reviewer 2 responded positively to the maps. 

The following has been added to the Discussion to acknowledge the disadvantages of choropleth maps:

We displayed country results on choropleth world maps with ternary plots and ternary-balance color schemes. These provided a general comparison of family planning compositions across marital groups in only two maps. While efficient in terms of information per map, these displays do require greater effort to interpret on behalf of the reader. Moreover, as with standard choropleth maps, information about uncertainty is not included; the reader would have to consult the tables also provided to retrieve this.

In reviewing this aspect of the submission, we also decided to slightly simplify Figure 3.

Global average levels and trends were described. However, regional results were summarized for 2019 only. I think it would be valuable to also discuss regional trends.

We added text describing main regional trends in the Results section.

Reviewer #2: 

This is an excellent paper that assembles lots of data and presents a careful analysis. There will be lots of interest in the results and the methods to arrive at those results. I have only a few minor issues to address.

Thank you for these comments. 

1. Lines 223-230. I could not understand the differences between the first sentence of the paragraph ‘For each country, the model generates estimates of …’ and the third sentence ‘In the modeling process…’ It seems to me that both sentences mean the same.

Apologies this was a typographical error. The third sentence now reads:

… an equivalent set of quantities is modelled: i) the proportion using a contraceptive method of any kind, ii) among those using any method of contraception, the proportion using any modern method the proportion using a modern contraceptive method, iii) among those not using any method of contraception, the proportion who have an unmet need, and iv) the proportion who have no need for family planning. 

2. Lines 280-281. The paper states that Group 0 is composed of countries with ‘very low’ rates of sexually activity among adolescent women, but ‘very low’ is never defined. Can you please provide the cut-off value used?

‘Very low’ was defined as the proportion of unmarried women of reproductive age who report sexual activity in past 28 days is below 2 percent. The data are obtained from recent DHS, MICS or other available surveys. This information is now presented in detail in “S1 Appendix”, an additional supplementary file added this revision. We modified the last part of Subsection “Hierarchical structures of models for married and unmarried women” of Section “Statistical Methods”; it now reads:

… Group 0 consisted of countries where sexual activity among unmarried women was estimated to be very low, defined as the proportion of unmarried women of reproductive age who report sexual activity in past 28 days was not more than 2 percent. Group 1 consisted of all other countries. To estimate the extent of sexual activity among unmarried women was estimated from the question on recent sexual activity in DHS, MICS and other available surveys. In countries without this data, we used information about the acceptance of sex between unmarried adults from other surveys and the level of religiosity. For further details on the data and methods used, see S1 Appendix.

We added more explanations in S1 Appendix, Subsection 1.2, including the text describing the data sources:

To account for the large differences in prevalence of sex among unmarried and not in a union women of reproductive age (UWRA) (Ueffing et al., 2019), two groups of countries were defined:

Sexual Activity Group 0 (SA0) Countries with very low levels of sexual activity, i.e., recent sexual activity (sexual intercourse in past four weeks) among UWRA was less than 2 percent;

Sexual Activity Group 1 (SA1) All other countries.

Countries were assigned to the same sexual activity group for the entire time period of estimation and projection.

Data on sexual activity among UWRA were available in 81 Demographic and Health Surveys (DHS) and Multiple Indicator Cluster Surveys (MICS). Countries with less than 2 percent sexually active (defined as having a sexual intercourse in past 28 days) among UWRA were assigned to Group 0, the rest to Group 1. Other data were used as proxies for countries with neither a DHS nor a MICS. For 43 countries, we used information on the acceptance of sex between unmarried adults from Pew Research Center (2014) (29 countries) and Inglehart et al. (2013) (14 countries). We used the most recently available data for each county. The remaining 71 countries, all in Asia and Northern Africa, were assigned on the basis of the proportion religious in the population using information from Pew Research Center (2012). 

This classification was done using information about sexual activity among UWRA, that is aged 15–49 years. We compared it to a classification based on sexual activity among women aged 15–19 years for the 81 countries with data from DHS or MICS. A small number of countries switched groups but these were countries with enough data on the family planning indicators such that their sexual activity grouping would have minimal impact on the posterior estimates. Given this finding, and the fact that we did not have information for the remaining 114 countries, we retained the classification based on sexual activity among UWRA. 

3. Line 369. ‘Compositional indicators were formed from the posterior medians…’ You cannot expect the medians of the four groups to sum to one (unlike the averages) so how did you derive the compositional indicators? Did you just normalize the medians?

Yes, we essentially normalized the medians so that a pre-determined set of identities held. We added text to note this in Subsection “Parameter Estimation and Software” of Section “Statistical Methods”. The modified paragraph now reads:

These MCMC samples were used to generate summary statistics for any parameter of interest. For example, the posterior median contraceptive prevalence, which was used as the measure of central tendency, was obtained by taking the empirical median of the marginal MCMC sample for that parameter. We followed Kantorová et al. [22, S1 Appendix, Subsection 3.7.6] and used small adjustments to medians only to ensure that basic identities held, such as that the proportion using any method equal the sum of the proportions using modern and traditional methods.

4. Line 423. I did not understand this sentence: ‘In these two regions, the estimated level of sexual activity among unmarried women in general, and among unmarried adolescents in particular, is low, accounting for the low levels of sexual activity, need for family planning and use of contraception.’ Perhaps you just need to remove ‘sexual activity’ from the second part of the sentence to avoid the tautology.

Thank you, we have removed ‘sexual activity’ exactly as suggested. 

5. At first glance, I thought the ternary diagrams and colored maps where going to be very hard to interpret. But they actually worked reasonably well. This is an interesting approach to presenting three-dimensional information.

Thank you for these comments.

6. I found it a bit cumbersome to think of the third component of the compositional indicator as ‘no need’ since it is not apparent to me how it should vary by region. But if I thought of it as ‘low levels of sexual activity’ then it was easy to follow. I assume that most of the ‘no need’ category is composed of women who are not sexually active, but there must be some component of those who want another child now. Could you show the proportion of the ‘no need’ group that comes from each component? It would certainly help to understand the indicator better.

It is correct that for the 15-19 age group the majority of ‘no need’ is explained by not being sexually active (in the last 4 weeks). The proportion is larger than 85% in all surveys and median at 98% (see figure below). Intended pregnancies and infecundity are more important in later age groups, but sexual activity remains most important factor for ‘no need’ group.

To clarify the diversity of the group of “not in need” we add a text in Definitions:

Women who are not in need of family planning are women who are not contraceptive users and are not classified as having an unmet need for family planning. For unmarried women, this group includes women who are either sexually inactive (as defined by no sexual intercourse in past 28 days), or sexually active who want to have a child, are infecund or are pregnant or postpartum amenorrhoeic (when the pregnancy was intended). Most unmarried women aged 15-19 years who have no need for family planning are sexually inactive.

Additional Changes

1. The supplementary document “S1 APPENDIX: Supplementary Figures” (filename ‘S1_Figures.pdf’) has been split into two : “S1 Appendix” and “S2 Appendix (Supplementary Figures)”. 

2. Provided names for supplementary information files.

3. Formatted references.

4. Simplified Figure 6.

5. Added sources for the base map.

---

## [Decision Letter · Decision Letter 1]

9 Feb 2021

Contraceptive Use and Needs among Adolescent Women Aged 15-19:  

Regional and Global Estimates and Projections from 1990 to 2030 from a Bayesian Hierarchical Modelling Study

PONE-D-20-16904R1

Dear Dr. Kantorová,

We’re pleased to inform you that your manuscript has been judged scientifically suitable for publication and will be formally accepted for publication once it meets all outstanding technical requirements.

Kind regards,

Philip Anglewicz, PhD

Academic Editor

PLOS ONE

Additional Editor Comments (optional):

Reviewers' comments:

Reviewer's Responses to Questions

**Comments to the Author**

1. If the authors have adequately addressed your comments raised in a previous round of review and you feel that this manuscript is now acceptable for publication, you may indicate that here to bypass the “Comments to the Author” section, enter your conflict of interest statement in the “Confidential to Editor” section, and submit your "Accept" recommendation.

Reviewer #1: All comments have been addressed

Reviewer #2: All comments have been addressed

2. Is the manuscript technically sound, and do the data support the conclusions?

Reviewer #1: Yes

Reviewer #2: Yes

3. Has the statistical analysis been performed appropriately and rigorously? 

Reviewer #1: Yes

Reviewer #2: Yes

4. Have the authors made all data underlying the findings in their manuscript fully available?

Reviewer #1: Yes

Reviewer #2: Yes

5. Is the manuscript presented in an intelligible fashion and written in standard English?

Reviewer #1: Yes

Reviewer #2: Yes

6. Review Comments to the Author

Reviewer #1: This article address is an important topic of interest to researchers, policymakers and advocates. It uses a sophisticated methodology, and also represents rigorous and comprehensive data processing. I greatly appreciate the additional information the authors have included, especially in the appendices. I think that the authors have done a nice job of responding to the comments, and I think the manuscript should be accepted.

One additional note:

The authors write:

"Out-of-sample validation exercises were performed to assess model fit; these are reported in S1 Appendix. The exercises indicated that the model fitted the data well and had good out-of-sample predictive validity"

Looking at Table B in the appendix, I see that the uncertainty intervals appear well calibrated, but the odds of under and over estimation different are somewhat different. I think it would be more accurate to state that the UI's are well-calibrated, and something along the lines that there is some evidence that the point estimates are more likely to be on one side than the other but the average errors are small on an absolute scale. It may also be useful to look into whether this is occurring in particular regions to understand this a little bit better.

Reviewer #2: (No Response)

7. PLOS authors have the option to publish the peer review history of their article (what does this mean?). If published, this will include your full peer review and any attached files.

Reviewer #1: No

Reviewer #2: No

---

## [Editor Report · Acceptance letter]

18 Feb 2021

PONE-D-20-16904R1 

Contraceptive Use and Needs among Adolescent Women Aged 15-19:Regional and Global Estimates and Projections from 1990 to 2030 from a Bayesian Hierarchical Modelling Study 

Dear Dr. Kantorová:

I'm pleased to inform you that your manuscript has been deemed suitable for publication in PLOS ONE. Congratulations! Your manuscript is now with our production department. 

Kind regards, 

on behalf of

Associate Professor Philip Anglewicz 

Academic Editor

PLOS ONE